# How to Train Your HiPPO: State Space Models with Generalized Orthogonal Basis Projections

## Abstract

Linear time-invariant state space models (SSM) are a classical model from engineering and statistics, that have recently been shown to be very promising in machine learning through the Structured State Space sequence model (S4). A core component of S4 involves initializing the SSM state matrix to a particular matrix called a HiPPO matrix, which was empirically important for S4's ability to handle long sequences. However, the specific matrix that S4 uses was actually derived in previous work for a particular *time-varying* dynamical system, and the use of this matrix as a *time-invariant* SSM had no known mathematical interpretation. Consequently, the theoretical mechanism by which S4 models long-range dependencies actually remains unexplained. We derive a more general and intuitive formulation of the HiPPO framework, which provides a simple mathematical interpretation of S4 as a decomposition onto exponentially-warped Legendre polynomials, explaining its ability to capture long dependencies. Our generalization introduces a theoretically rich class of SSMs that also lets us derive more intuitive S4 variants for other bases such as the Fourier basis, and explains other aspects of training S4, such as how to initialize the important timescale parameter. These insights improve S4's performance to 86% on the Long Range Arena benchmark, with 96% on the most difficult Path-X task.

## 1 Introduction

The Structured State Space model (S4) is a recent deep learning model based on continuous-time dynamical systems that has shown promise on a wide variety of sequence modeling tasks (Gu et al., 2022a). It is defined as a particular linear time-invariant (LTI) state space model (SSM), which give it multiple properties (Gu et al., 2021): as an SSM, S4 can be simulated as a discrete-time recurrence for efficiency in online or autoregressive settings, and as a LTI model, S4 can be converted into a convolution for parallelizability and computational efficiency at training time. These properties give S4 remarkable computational efficiency and performance, especially when modeling continuous signal data and long sequences.

Despite its potential, several aspects of the S4 model remain poorly understood. Most notably, Gu et al. (2022a) claim that the long range abilities of S4 arise from instantiating it with a particular "**HiPPO matrix**" (Gu et al., 2020). However, this matrix was actually derived in prior work for a different (*time-varying*) setting, and the use of this matrix in S4 (a *time-invariant* SSM) did not have a mathematical interpretation. Consequently, the mechanism by which S4 truly models long-range dependencies is actually not known. Beyond this initialization, several other aspects of parameterizing and training S4 remain poorly understood. For example, S4 involves an important timescale parameter $\Delta$, and suggests a method for parameterizing and initializing this parameter, but does not discuss its meaning or provide a justification.

This work aims to provide a comprehensive theoretical exposition of several aspects of S4. The major contribution of this work is a cleaner, more intuitive, and much more general formulation of the HiPPO framework. This result directly generalizes all previous known results in this line of work (Voelker et al., 2019; Gu et al., 2020; 2021; 2022a). As immediate consequences of this framework:

- We prove a theoretical interpretation of S4's state matrix $\boldsymbol{A}$, explaining S4's ability to capture long-range dependencies via decomposing the input with respect to an infinitely long, exponentially-decaying measure.
- We derive new HiPPO matrices and corresponding S4 variants that generalize other nice basis functions. For example, our new method S4-FouT produces *truncated Fourier basis functions*. This method thus automatically captures sliding Fourier transforms (e.g. the STFT and spectrograms), which are ubiquitous as a hand-crafted signal processing tool, and can also represent any *local convolution*, thus generalizing conventional CNNs.
- We provide an intuitive explanation of the timescale $\Delta$, which has a precise interpretation as controlling the length of dependencies that the model captures. Our framework makes it transparent how to initialize $\Delta$ for a given task, as well as how to initialize the other parameters (in particular, the last SSM parameter $\boldsymbol{C}$) to make a deep SSM variance-preserving and stable.

Empirically, we validate our theory on synthetic function reconstruction and memorization tasks, showing that empirical performance of state space models in several settings is predicted by the theory. For example, our new S4-FouT method, which can provably encode a spike function as its convolution kernel, performs best on a continuous memorization task compared to other SSMs and other models, when $\Delta$ is initialized correctly. Finally, we show that the original S4 method is still best on very long range dependencies, achieving a new state of the art of **86%** average on Long Range Arena, with **96%** on the most difficult Path-X task that even the other SSM variants struggle with.

## 2 FRAMEWORK

We present our improved framework for state space models and online reconstruction of signals. Section 2.1 discusses background on SSMs, including their connection to convolutions for time-invariant systems. Section 2.2 defines new subclasses of SSMs with special properties that can be used for online function reconstruction, simplifying and generalizing the original HiPPO framework. An extended background and related work section can be found in Appendix A.

### 2.1 STATE SPACE MODELS: A CONTINUOUS-TIME LATENT STATE MODEL

The state space model (**SSM**) is defined by the differential equation (1) and (2). Given an input sequence $u$ of length $N$, it maps a 1-D input signal $u(t)$ to an $N$-D latent state $x(t)$ before projecting to a 1-D output signal $y(t)$.

$$
\begin{aligned}
x'(t) &= \boldsymbol{A}(t)x(t) + \boldsymbol{B}(t)u(t) \quad\quad (1) & K(t) &= \boldsymbol{C}e^{t\boldsymbol{A}}\boldsymbol{B} \\
y(t) &= \boldsymbol{C}(t)x(t) + \boldsymbol{D}(t)u(t) \quad\quad (2) & y(t) &= (K * u)(t)
\end{aligned} \quad (3)
$$

We will generally assume $\boldsymbol{D} = 0 \in \mathbb{R}$ and omit it for simplicity, unless explicitly mentioned.

SSMs can in general have dynamics that change over time, i.e. the matrix $\boldsymbol{A} \in \mathbb{R}^{N \times N}$, and vectors $\boldsymbol{B} \in \mathbb{R}^{N \times 1}, \boldsymbol{C} \in \mathbb{R}^{1 \times N}$ are a function of $t$ in (1) and (2). However, when they are constant the system is **linear time invariant (LTI)**, and is equivalent to a convolutional system (3). The function $K(t)$ is called the **impulse response** which can also be defined as the output of the system when the input $u(t) = \delta(t)$ is the impulse or Dirac delta function. We will call these **time-invariant state space models (TSSM)**. These are particularly important because the equivalence to a convolution makes TSSMs parallelizable and very fast to compute, which is critical for S4's efficiency.

Our treatment of SSMs will consider the $(\boldsymbol{A}, \boldsymbol{B})$ parameters separately from $\boldsymbol{C}$. We will refer to an SSM as either the tuple $(\boldsymbol{A}, \boldsymbol{B}, \boldsymbol{C})$ (referring to (3)) or $(\boldsymbol{A}, \boldsymbol{B})$ (referring to Definition 1) when the context is unambiguous. We also drop the T in TSSM when the context is clearly time-invariant.

**Definition 1.** *Given a TSSM $(\boldsymbol{A}, \boldsymbol{B})$, $e^{t\boldsymbol{A}}\boldsymbol{B}$ is a vector of $N$ functions which we call the **SSM basis**. The individual basis functions are denoted $K_n(t) = \boldsymbol{e}_n^\top e^{t\boldsymbol{A}}\boldsymbol{B}$, which satisfy $x_n(t) = (u * K_n)(t) = \int_{-\infty}^{t} K_n(t-s)u(s)ds$. Here $\boldsymbol{e}_n$ is the one-hot basis vector.*

This definition is motivated by noting that the SSM convolutional kernel is a linear combination of the SSM basis controlled by the vector of coefficients $\boldsymbol{C}$, $K(t) = \sum_{n=0}^{N-1} \boldsymbol{C}_n K_n(t)$.

We note that Definition 1 has not appeared in prior works on deep SSMs, but is a new perspective taken by this work for understanding and visualizing these models.

**Discrete SSM with Timescales.** To be applied on a discrete input sequence $(u_0, u_1, ...)$ instead of continuous function $u(t)$, (1) must be discretized by a **step size** $\Delta$ that represents the resolution of the input. A poorly understood question from prior work is how to interpret and choose this $\Delta$ parameter, especially when the input $u_k$ does not actually arise from uniformly sampling an underlying continuous signal. S4 specifies to log-uniformly initialize $\Delta$ in the range $(\Delta_{min}, \Delta_{max}) = (0.001, 0.1)$, but does not provide a concrete justification. In Section 3.3 we show a simpler interpretation of $\Delta$ directly in terms of the length of dependencies in a discrete input sequence.

## 2.2 HiPPO: High-order Polynomial Projection Operators

S4 is defined as a TSSM where $(\boldsymbol{A}, \boldsymbol{B})$ is initialized with a particular formula (4). This was called the HiPPO matrix in (Gu et al., 2022a), but is actually just one of several such special matrices derived in (Gu et al., 2020). To disambiguate other variants of S4, we refer to the full S4 method using this HiPPO SSM as **S4-LegS**. Other cases considered in this work include LegT from prior work (5) and FouT that we introduce in this work (6).

**(HiPPO-LegS)**

$$\boldsymbol{A}_{nk} = -(2n+1)^{\frac{1}{2}}(2k+1)^{\frac{1}{2}} \cdot \begin{cases} 1 & n > k \\ \frac{n+1}{2n+1} & n = k \\ 0 & n < k \end{cases} \quad (4)$$

$$\boldsymbol{B}_n = (2n+1)^{\frac{1}{2}}$$

**(HiPPO-LegT)**

$$\boldsymbol{A}_{nk} = -(2n+1)^{\frac{1}{2}}(2k+1)^{\frac{1}{2}} \cdot \begin{cases} 1 & k \le n \\ (-1)^{n-k} & k \ge n \end{cases} \quad (5)$$

$$\boldsymbol{B}_n = (2n+1)^{\frac{1}{2}}$$

**(HiPPO-FouT)**

$$\boldsymbol{A}_{nk} = \begin{cases} -2 & n = k = 0 \\ -2\sqrt{2} & n = 0, k \text{ odd} \\ -2\sqrt{2} & k = 0, n \text{ odd} \\ -4 & n, k \text{ odd} \\ 2\pi k & n - k = 1, k \text{ odd} \\ -2\pi n & k - n = 1, n \text{ odd} \\ 0 & \text{otherwise} \end{cases} \quad (6)$$

$$\boldsymbol{B}_n = \begin{cases} 2 & n = 0 \\ 2\sqrt{2} & n \text{ odd} \\ 0 & \text{otherwise} \end{cases}$$

These matrices were originally motivated by the question of "online memorization" of an input signal. In the following, we present an improved version of the HiPPO framework that addresses this problem.

The key idea is that for a suitably chosen SSM basis $\boldsymbol{A}, \boldsymbol{B}$, at any time $t$, the current state $x(t)$ can be used to approximately reconstruct the entire input $u$ up to time $t$. In Appendix A.1, we describe the full HiPPO framework as described in (Gu et al., 2020). In particular, suppose that the basis functions satisfy Definition 2.

**Definition 2.** *We call an SSM $(\boldsymbol{A}(t), \boldsymbol{B}(t))$ an **orthogonal SSM (OSSM)** for the basis $p_n(t,s)$ and measure $\omega(t,s) \ge 0$ if the functions $K_n(t,s) = p_n(t,s)\omega(t,s)$ satisfy, at all times $t$,*

$$x_n(t) = \int_{-\infty}^{t} K_n(t,s)u(s)ds \qquad \int_{-\infty}^{t} p_n(t,s)p_m(t,s)\omega(t,s)ds = \delta_{n,m}. \quad (7)$$

*In the case of a **time-invariant OSSM (TOSSM)**, $K_n(t,s) =: K_n(t-s)$ (depends only on $t-s$), giving us Definition 1 with measure $\omega(t-s) := \omega(t,s)$ and basis $p_n(t-s) := p_n(t,s)$.*

To be more specific about terminology, $p_n$ and $\omega_n$ are called the *basis* and *measure* for *orthogonal* SSMs (Definition 2), while $K_n$ are called the *SSM basis kernels* which applies more generally to all SSMs (Definition 1). The distinction will be made clear from context, notation, and the word "kernel" referring to $K_n$. Note that for OSSMs, $(p_n, \omega_n)$ and $K_n$ are uniquely determined by each other (Proposition 6 in Appendix C.2), so we can refer to an OSSM by either.

Defining $p_n^{(t)}(s) = p_n(t,s)$ and similarly $\omega^{(t)}(s) = \omega(t,s)$ for every fixed $t$, the bases $p_n^{(t)}$ are orthonormal in the Hilbert space with inner product $\langle p,q \rangle = \int p(s)q(s)\omega^{(t)}(s)ds$. By equation (7), we have $x_n(t) = \int_{-\infty}^{t} u(s)K_n(t,s)ds = \langle u, p_n^{(t)} \rangle_{\omega^{(t)}}$. Thus at all times $t$, the state vector $x(t)$ is simply *the projections of $u|_{\le t}$ onto a orthonormal basis*, so that $u$ can be reconstructed from $x(t)$. In the HiPPO framework, this reconstruction is called the **online function approximation** problem (Gu et al., 2020).

**Proposition 1.** *Consider an OSSM that satisfies* (7) *and suppose that in the limit* $N \to \infty$, *for a fixed time* $t$, *the* $p_n^{(t)}$ *are complete on the support of* $\omega$. *Then* $u(s) = \sum_{n=0}^{\infty} x_n(t) p_n(t,s)$ *for all* $s \leq t$.

HiPPO can thus be viewed as a framework for deriving specific SSMs that do satisfy (7). The original HiPPO methods and its generalizations (Gu et al., 2020; 2021) primarily focused on the case when the $p_n$ are orthogonal polynomials, and specifically looked for solutions to (7), which turn out to be SSMs. We have rephrased the HiPPO definition in Definition 2 to start directly from SSMs and hence is more general. (See Appendix A.1 for an overview of the original HiPPO setup.) We discuss the two most important cases previously introduced.

**HiPPO-LegT.** (5) is a TOSSM that approximates the Legendre polynomials.

**Definition 3.** *Let* $\mathbb{I}(t)$ *be the indicator function for the unit interval* [0,1]. *Let* $L_n(t)$ *be the Legendre polynomials rescaled to be orthonormal on* [0,1], *i.e.,* $\int L_n(t) L_m(t) \mathbb{I}(t) dt = \delta_{n,m}$.

**Proposition 2.** *As* $N \to \infty$, *the SSM* (5) *is a TOSSM with* $\omega(t) = \mathbb{I}(t)$, $p_n(t) = L_n(t)$.

This particular system was the precursor to HiPPO and has also been variously called the Legendre Delay Network (LDN) or Legendre Memory Unit (LMU) (Voelker, 2019; Voelker et al., 2019). The original motivation of this system was not through the online function approximation formulation of HiPPO, but through finding an optimal SSM approximation to the **delay network** that has impulse response $K(t) = \delta(t-1)$ representing a time-lagged output by 1 time unit. This is visualized in Appendix C.4.4 Fig. 7. We state and provide an alternate proof of this result in Appendix C.4.4, Theorem 10.

**HiPPO-LegS.** Unlike the HiPPO-LegT case, which is an LTI system (1) (i.e. TOSSM), the HiPPO-LegS matrix (4) was *meant to be used in a time-varying system* $x'(t) = \frac{1}{t} \boldsymbol{A} x(t) + \frac{1}{t} \boldsymbol{B} u(t)$ (Gu et al., 2020). In contrast to HiPPO-LegT, which reconstructs onto the truncated Legendre polynomials in sliding windows $[t-1,t]$, HiPPO-LegS reconstructs onto Legendre polynomials on "scaled" windows $[0,t]$; since the window changes across time, the system is not time-invariant. Specifically, we have:

**Theorem 3.** *The SSM* $(\frac{1}{t} \boldsymbol{A}, \frac{1}{t} \boldsymbol{B})$ *for* $(\boldsymbol{A}, \boldsymbol{B})$ *in* (4) *is an OSSM with*

$$\omega(t,s) = \frac{1}{t} \cdot \mathbb{I}(s/t) \qquad\qquad p_n(t,s) = L_n(s/t).$$

However, the S4 model applies the exact same formula (4) inside the *time-invariant* SSM (1), i.e. dropped the $\frac{1}{t}$ term, which had no mathematical interpretation (see Appendix A.1 for more details). In other words, while $(\frac{1}{t} \boldsymbol{A}, \frac{1}{t} \boldsymbol{B})$ is an OSSM, it was not known whether the TSSM $(\boldsymbol{A}, \boldsymbol{B})$ is a TOSSM. Given that the performance of SSM models is very sensitive to these matrices $\boldsymbol{A}$ (Gu et al., 2022a; Gupta, 2022), it remained a mystery why this works. In Section 3 we will prove that (4) actually does correspond to a TOSSM.

**Naming convention.** We use HiPPO-[SSM] to refer to a fixed OSSM $(\boldsymbol{A}, \boldsymbol{B})$ suitable for online function approximation, where [SSM] is a suffix (e.g. LegS, LegT) that abbreviates the corresponding basis functions (e.g. scaled Legendre, truncated Legendre). S4-[SSM] refers to the corresponding trainable layer $(\boldsymbol{A}, \boldsymbol{B}, \boldsymbol{C})$ with randomly initialized $\boldsymbol{C}$, trained with S4's representation and computational algorithm (Gu et al., 2022a).

**Other SSMs.** Several variants of S4 have been introduced, including several simpler diagonal SSMs (DSS (Gupta, 2022), S4D (Gu et al., 2022b), S5 (Smith et al., 2022)). Notably, these methods are all based on approximations of HiPPO-LegS, and our new theory explains why they perform well (Gu et al., 2022b). However, they are not OSSMs, and in Section 4 we show several settings where the full S4 variants based on OSSMs outperform these variants.

## 3 Generalized HiPPO: General Orthogonal Basis Projections

In Section 3.1, we prove that the LTI HiPPO-LegS is actually a TOSSM and show closed formulas for its basis functions. In Section 3.2, we include more specific results on finite-window SSMs, including introducing a new method HiPPO-FouT based on truncated Fourier functions, and proving previously established conjectures. Section 3.3 shows more general properties of TOSSMs, which establish guidelines for interpreting and initializing SSM parameters such as the timescale $\Delta$.

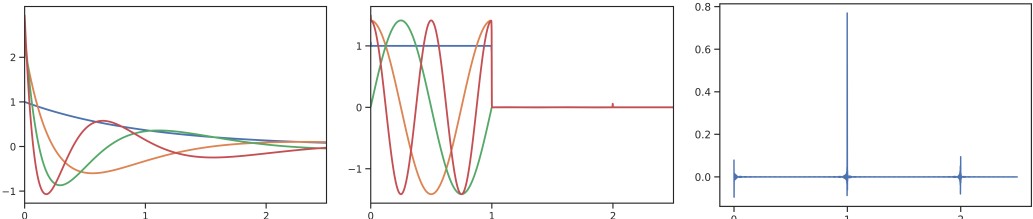

Figure 1: (*Left*: **LegS**) We prove that the particular $\boldsymbol{A}$ matrix chosen in S4 produces Legendre polynomials under an exponential re-scaling, resulting in smooth basis functions with a closed form formula. (*Middle, Right*: **FouT**) We derive a new SSM that produces approximations to the **truncated Fourier** basis, perhaps the most intuitive and ubiquitous set of basis functions. This method generalizes sliding Fourier Transforms and local convolutions (i.e. CNNs), and can also encode spike functions to solve classic memorization tasks.

Our main, fully general, result is Theorem 8 in Appendix C.2, which describes a very general way to derive OSSMs for various SSM basis functions $K_n(t,s)$. This result can be instantiated in many ways to generalize all previous results in this line of work.

### 3.1 EXPLANATION OF S4-LEGS

We showcase the generality of Theorem 8 by stating the following special case containing a sub-class of time varying OSSMs (which are themselves rich enough to explain both S4-LegS and HiPPO-LegS):

**Corollary 3.1.** *Define $\sigma(t,s) = \exp(a(s) - a(t))$ for any differentiable function $a$. The SSM $(a'(t)\boldsymbol{A}, a'(t)\boldsymbol{B})$ is an OSSM with*
$$\omega(t,s) = \mathbb{I}(\sigma(t,s))a'(s)\sigma(t,s) \qquad p_n(t,s) = L_n(\sigma(t,s)).$$

We show the matrices $(\boldsymbol{A}, \boldsymbol{B})$ in (4) are deeply related to the Legendre polynomials $L_n$ defined in Definition 3. In particular, as more specific corollaries of Corollary 3.1, we recover both the original time-varying interpretation of the matrix in (4), as well as the instantiation of LegS as a time-invariant system. If we set $a'(t) = \frac{1}{t}$, then we recover the scale-invariant HiPPO-LegS OSSM in Theorem 3:

**Corollary 3.2** (Scale-Invariant HiPPO-LegS, Theorem 3)**.** *The SSM $(\frac{1}{t}\boldsymbol{A}, \frac{1}{t}\boldsymbol{B})$ is a TOSSM for basis functions $K_n(t) = \frac{s}{t}L_n(\frac{s}{t})$ and measure $\omega = \frac{1}{t}\mathbb{I}[0,1]$ where $\boldsymbol{A}$ and $\boldsymbol{B}$ are defined as in (4).*

And if we set $a'(t) = 1$, this shows a new result for the time-invariant HiPPO-LegS TOSSM:

**Corollary 3.3** (Time-Invariant HiPPO-LegS)**.** *The SSM $(\boldsymbol{A}, \boldsymbol{B})$ is a TOSSM with*
$$\omega(t) = e^{-t} \qquad p_n(t) = L_n(e^{-t}).$$

This explains why removing the $\frac{1}{t}$ factor from HiPPO-LegS still works: it is orthogonalizing onto the Legendre polynomials with an exponential "warping" or change of basis on the time axis (Fig. 1,(*Left*)).

### 3.2 FINITE WINDOW TIME-INVARIANT ORTHOGONAL SSMs

For the remainder of this section, we restrict to the time-invariant SSM setting (3). A second important instantiation of Theorem 8 covers cases with a discontinuity in the SSM basis functions $K_n(t)$, which requires infinite-dimensional SSMs to represent. The most important type of discontinuity occurs when $K_n(t)$ is supported on a finite window, so that these TSSMs represent sliding window transforms.

#### 3.2.1 S4-FOUT

Using the more general framework (Theorem 8) that does not necessarily require polynomials as basis functions, we derive a TOSSM that projects onto **truncated Fourier functions**.

**Theorem 4.** *As $N \to \infty$, the SSM for (6) is a TOSSM with $\omega(t) = \mathbb{I}(t)$, and $\{p_n\}_{n \geq 1}$ are the truncated Fourier basis functions orthonormal on $[0,1]$, ordered in the form $\{p_n\}_{n \geq 0} = (1, c_0(t), s_0(t), ...)$, where $s_m(t) = \sqrt{2}\sin(2\pi mt)$ and $c_m(t) = \sqrt{2}\cos(2\pi mt)$ for $m = 0, ..., N/2$.*

This SSM corresponds to Fourier series decompositions, a ubiquitous tool in signal processing, but represented as a state space model. The basis is visualized in Fig. 1 (middle) for state size $N = 1024$.

A benefit of using these well-behaved basis functions is that we can leverage classic results from Fourier analysis. For example, it is clear that taking linear combinations of the truncated Fourier basis can represent any function on $[0,1]$, and thus S4-FouT can represent any local convolution (i.e. the layers of modern CNNs) (cf. Theorem 9 in Appendix C.4).

### 3.2.2 Approximating Delay Networks

An interesting property of these finite window TSSMs is that they can approximate **delay functions**. This is defined as a system with impulse response $K(t) = \delta(t-1)$. Any HiPPO method involving finite windows should have this capability, in particular, the finite window methods LegT and FouT:

**Theorem 5.** *For the FouT system $\boldsymbol{A}$ and $\boldsymbol{B}$ (6), let $\boldsymbol{C}$ be (twice) the vector of evaluations of the basis functions $\boldsymbol{C}_n = 2 \cdot p_n(1)$ and let $\boldsymbol{D} = 1$. For the LegT system $\boldsymbol{A}$ and $\boldsymbol{B}$ (5), let $\boldsymbol{C}$ be the vector of evaluations of the basis functions $\boldsymbol{C}_n = p_n(1) = (1+2n)^{\frac{1}{2}} (-1)^n$ and let $\boldsymbol{D} = 0$.*

*Then the SSM kernel $K(t) = \boldsymbol{C} e^{t\boldsymbol{A}} \boldsymbol{B} + \boldsymbol{D}\delta(t)$ limits to $K(t) \to \delta(t-1)$ as $N \to \infty$.*

Theorem 5 is visualized in Fig. 1 for FouT, and Fig. 7 in Appendix C.4. Further, the result for LegT can be characterized even more tightly for finite $N$ (cf. Theorem 10 in Appendix C.4). The above result provides theoretical justification for why S4-FouT excels at dense memorization tasks (see Section 4).

### 3.3 Properties of Time-invariant Orthogonal SSMs

We describe several general properties of TOSSMs, which let us answer the following questions: What does $\Delta$ intuitively represent, and how should it be set in an SSM model? So far, this had been done in an ad-hoc way. It turns out that for TOSSMs, these two questions are closely related and have intuitive interpretations. See Appendix C.5 for more details on other properties of TOSSMs related to their closure and normalization.

**Timescales.** As discussed in Section 2, converting from continuous to discrete time involves a parameter $\Delta$ that represents the step size of the discretization. This is an unintuitive quantity when working directly with discrete data, especially if it is not sampled from an underlying continuous process.

We observe the following fact: for all standard discretization methods (e.g. Euler, backward Euler, generalized bilinear transform, zero-order hold (Gu et al., 2021)), the discretized system depends on $(\boldsymbol{A},\boldsymbol{B})$, and $\Delta$ *only through their products* $(\Delta\boldsymbol{A},\Delta\boldsymbol{B})$. This implies that the SSM $(\boldsymbol{A},\boldsymbol{B})$ discretized at step size $\Delta$ is computationally equivalent to the SSM $(\Delta\boldsymbol{A},\Delta\boldsymbol{B})$ discretized at step size 1.

Therefore, $\Delta$ can be viewed just as a scalar scaling of the base SSM instead of changing the rate of the input. In the context of TOSSMs, this just stretches the underlying basis and measure (Scalar Scaling).

The most intuitive example of this is for a finite window TOSSM such as LegT or FouT. Discretizing this system with step size $\Delta$ is equivalent to considering the system $(\Delta\boldsymbol{A},\Delta\boldsymbol{B})$ with step size 1, which produces basis functions supported exactly on $[0,\frac{1}{\Delta}]$. The interpretation of the timescale $\Delta$ lends to simple discrete-time corollaries of the previous continuous-time results. For example, LegT and FouT *represent sliding windows of* $1/\Delta$ *elements* in discrete time.

**Corollary 3.4.** *By Theorem 5, as $N \to \infty$, the discrete convolutional kernel $\overline{\boldsymbol{K}} \to \boldsymbol{e}_{\lceil \Delta^{-1} \rceil}$, i.e. the discrete delay network with lag $\frac{1}{\Delta}$.*

**Corollary 3.5.** *For HiPPO-FouT matrices $(\boldsymbol{A},\boldsymbol{B})$, by Theorem 4, as $N \to \infty$, the discrete convolutional kernel $\overline{\boldsymbol{K}}$ (over the choice of $\boldsymbol{C}$) can represent any local convolution of length $\lfloor \Delta^{-1} \rfloor$.*

This discussion motivates the following definition. Properly normalized TOSSMs $(\boldsymbol{A},\boldsymbol{B})$ will model dependencies of expected length 1, and $\Delta$ modulates it to model dependencies of length $\frac{1}{\Delta}$, allowing fine-grained control of the context size of a TOSSM.

Table 1: (**Long Range Arena**) Accuracy (std.) on full suite of LRA tasks. Hyperparameters in Appendix B. ✗ denotes failure to learn better than random guessing, following convention from Tay et al. (2021); Gu et al. (2022a).

| MODEL | LISTOPS | TEXT | RETRIEVAL | IMAGE | PATHFINDER | PATH-X | AVG |
|---|---|---|---|---|---|---|---|
| S4-LegS | 59.60 (0.07) | **86.82** (0.13) | **90.90** (0.15) | 88.65 (0.23) | 94.20 (0.25) | **96.35** (-) | **86.09** |
| S4-FouT | 57.88 (1.90) | 86.34 (0.31) | 89.66 (0.88) | **89.07** (0.19) | **94.46** (0.24) | ✗ | 77.90 |
| S4-LegS/FouT | **60.45** (0.75) | 86.78 (0.26) | 90.30 (0.28) | 89.00 (0.26) | 94.44 (0.08) | ✗ | 78.50 |
| S4 (original) | 58.35 | 76.02 | 87.09 | 87.26 | 86.05 | 88.10 | 80.48 |
| Transformer | 36.37 | 64.27 | 57.46 | 42.44 | 71.40 | ✗ | 53.66 |

**Definition 4** (Timescale of TOSSM). *Define* $\mathbb{E}[\omega] = \frac{\int_0^\infty t\omega(t)dt}{\int_0^\infty \omega(t)dt}$ *to be the* timescale *of a TOSSM having measure* $\omega(t)$. *A TOSSM is* timescale normalized *if it has timescale* 1.

By this definition, HiPPO-LegS is timescale normalized. This motivates S4's initialization of $\Delta$ log-uniformly in (0.001,0.1), covering a geometric range of sensible timescales (expected length 10 to 1000). In Section 4 we show that the timescale can be chosen more precisely when lengths of dependencies are known.

## 4    EXPERIMENTS

We study the empirical tradeoffs of our proposed S4 variants. We compare several S4 variants based on the TOSSMs introduced in this work, as well as to simpler diagonal SSMs called S4D that are not orthogonal SSMs (Gu et al., 2022b). Corresponding to our main contributions, we hypothesize that

- S4-LegS excels at sparse memorization tasks because it represents very smooth convolution kernels that memorize the input against an infinitely-long measure (Corollary 3.3, Fig. 1). Conversely, it is less appropriate at short-range tasks with dense information because it smooths out the signal.
- S4-FouT excels at dense memorization tasks because it can represent spike functions that pick out past elements at particular ranges (Section 3.2.2). However, it is less appropriate at very long range tasks because it represents a finite (local) window.
- $\Delta$ can be initialized precisely based on known time dependencies in a given task.

### 4.1    LONG RANGE ARENA

The Long Range Arena (LRA) benchmark is a suite of sequence classification tasks designed to stress test sequence models on modeling long sequences. We improve S4's previous state of the art by another 6 points (Table 1). Validating our hypothesis, S4-LegS is extremely strong at the hardest long-range task (Path-X) involving sparse dependencies of length 16384, which FouT cannot solve because it is a finite window method.

Compared to the original S4 model, the S4-LegS method in Table 1 is the same model but differs by improving some sensible hyperparameters; the main differences are (i) using a bidirectional instead of autoregressive model, since the tasks do not require causality (ii) adopting a more standard cosine learning rate scheduler rather than decaying on validation plateau, and (iii) increasing weight decay regularization.

On top of these general changes, the primary source of improvements on Path-X performance arises from applying the theory of timescales in Section 3.3. Fig. 2 illustrates the importance of setting $\Delta$ correctly. Instead of the standard initialization of $(\Delta_{min},\Delta_{max}){=}(0.001,0.1)$, these results were obtained by lowering the initialization of $\Delta$ by a factor of 10 in accordance with known length of dependencies in the Path-X task.

### 4.2    THEORY: FUNCTION RECONSTRUCTION, TIMESCALES, NORMALIZATION

Fig. 3 confirms the HiPPO theory of online function reconstruction (Proposition 1) for the proposed TOSSMs LegS and FouT. A followup question is whether the theory is *necessary* to develop methods that have this functionality, or whether other SSMs can still learn this

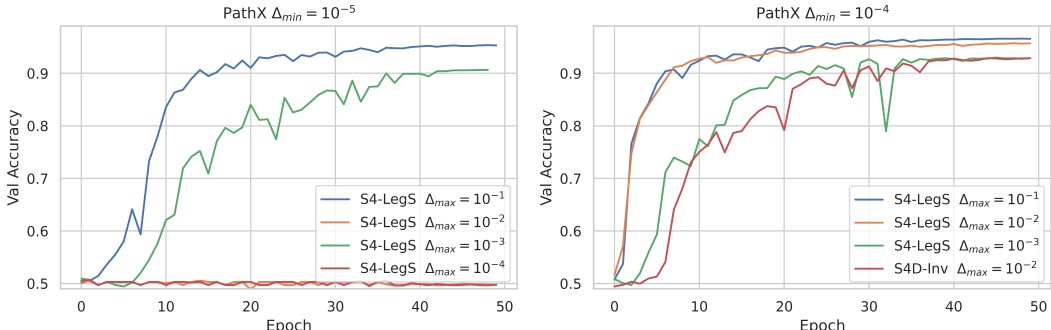

Figure 2: (**Validation curves on Path-X.**) (*Left*) Setting $\Delta_{min}$ too small can solve the task, but is inconsistent. (*Right*) A good setting of $\Delta_{min}$ can consistently solve the task. Note that the timescale of each feature is up to $\frac{1}{\Delta_{min}} = 10^4$, which is on the order of (but not exceeding) the length of the task $L = 16384$. Empirically, performance is best when spreading out the range of $\Delta$ with a larger $\Delta_{max}$ that covers a wider range of timescales and can potentially learn features at different resolutions, which are combined by a multi-layer deep neural network. We also show a diagonal variant of S4-LegS called S4D-Inv introduced in (Gu et al., 2022b) which approximates S4-LegS, but is still worse.

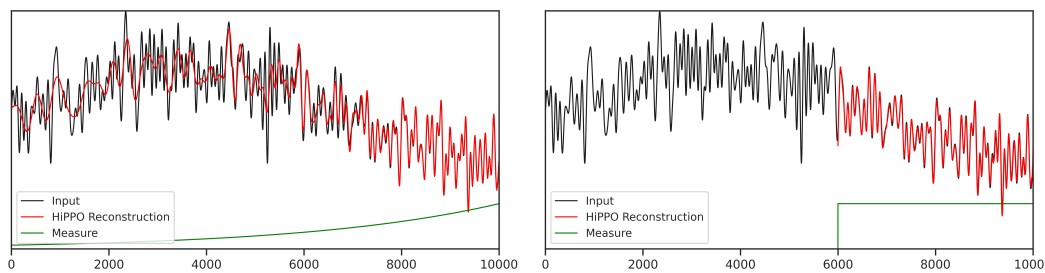

Figure 3: (**New HiPPO methods**) Function reconstruction predicted by our general theory. An input signal of length 10000 is processed sequentially, maintaining a state vector of size only $x(t) \in \mathbb{R}^{64}$, which is then used to approximately reconstruct the entire history of the input. (*Left*) HiPPO-LegS (as an LTI system) orthogonalizes on the Legendre polynomials warped by an exponential change of basis, smoothening them out. This basis is orthogonal with respect to an exponentially decaying measure. Matching the intuition, the reconstruction is very accurate for the recent past and degrades further out, but still maintains information about the full history of the input, endowing it with long-range modeling capacity. This is the same as S4. (*Right*) HiPPO-FouT orthogonalizes on the truncated Fourier basis, similar to the original HiPPO-LegT or LMU.

task when trained. Appendix B.1, Fig. 6 analyzes a synthetic **Reconstruction Task** against a uniform measure, which validates that S4-LegT and S4-FouT are far better than other SSM variants, particularly when $\Delta$ is initialized properly based on the length of the task.

### 4.3 MEMORIZATION: THE DELAY (CONTINUOUS COPYING) TASK

Next, we study how the synthetic reconstruction ability transfers to other tasks. The **Delay Task** requires models to learn a sequence-to-sequence map whose output is the input lagged by a fixed time period (Fig. 4a). For recurrent models, this task can be interpreted as requiring models to maintain a memory buffer that continually remembers the latest elements it sees. This capability was the original motivation for the Legendre Memory Unit, the predecessor to HiPPO-LegT, which was explicitly designed to solve this task because it can encode a spike kernel (Fig. 7). In Fig. 4b, we see that our new S4-FouT actually outperforms S4-LegT, which both outperform all other methods when the timescale $\Delta$ is set correctly. We note that this task with a lag of just 1000 time steps is too hard for baselines such as an LSTM and Transformer, which empirically did not learn better than random guessing (RMSE 0.43).

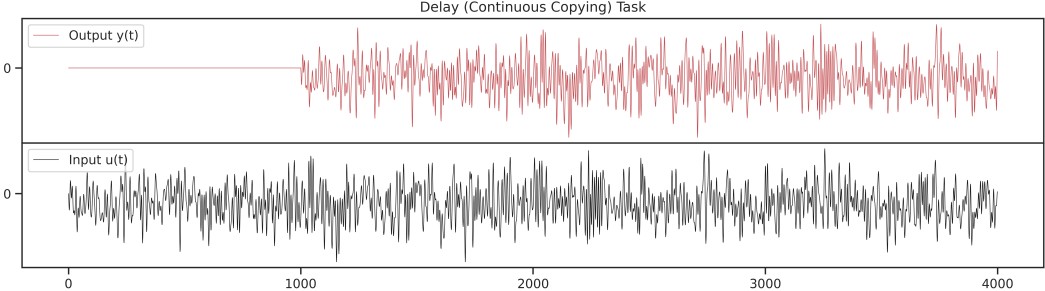

(a) (**Delay Task**) Models perform a mapping from $\mathbb{R}^{4000} \to \mathbb{R}^{4000}$ where the target output is lagged by 1000 steps, with error measured by RMSE. The input is a white noise signal bandlimited to $1000Hz$. We use single layer SSMs with state size $N = 1024$.

| (Δ min, Δ max) | Frozen (A, B) | Trainable (A, B) |
| --- | --- | --- |
| (5e-4, 5e-4) | 0.2891 | 0.2832 |
| (1e-3, 1e-3) | 0.1471 | 0.1414 |
| (2e-3, 2e-3) | 0.0584 | 0.0078 |
| (4e-3, 4e-3) | 0.4227 | 0.1262 |
| (8e-3, 8e-3) | 0.4330 | 0.2928 |
| (5e-4, 1e-1) | 0.2048 | 0.1537 |
| (1e-3, 1e-1) | 0.2017 | 0.1474 |
| (2e-3, 1e-1) | 0.3234 | 0.2262 |
| (4e-3, 1e-1) | 0.4313 | 0.3417 |
| (8e-3, 1e-1) | 0.4330 | 0.4026 |

| | (Δ min, Δ max) = (1e-3, 1e-1) | | (Δ min, Δ max) = (2e-3, 2e-3) | |
| --- | --- | --- | --- | --- |
| | Frozen (A, B) | Trainable (A, B) | Frozen (A, B) | Trainable (A, B) |
| S4-LegS | 0.3072 | 0.0379 | 0.2369 | 0.0130 |
| S4-LegT | 0.2599 | 0.1204 | 0.0226 | 0.0129 |
| S4-FouT | 0.2151 | 0.1474 | 0.0304 | 0.0078 |
| S4-LegS+FouT | 0.1804 | 0.0309 | 0.0250 | 0.0080 |
| S4D-LegS | 0.1378 | 0.0337 | 0.0466 | 0.0140 |
| S4D-Inv | 0.1489 | 0.0243 | 0.0605 | 0.0186 |
| S4D-Lin | 0.1876 | 0.1653 | 0.0421 | 0.0144 |

(b) (**RMSE**) (*Left*) Setting $\Delta$ appropriately makes a large difference. For FouT $(\boldsymbol{A},\boldsymbol{B})$, which encode *finite window* basis functions (Fig. 1), the model can see a history of length up to $\frac{2}{\Delta}$. For example, setting $\Delta$ too large means the model cannot see 1000 steps in the past, and performs at chance. Performance is best at the theoretically optimal value of $\Delta = 2 \cdot 10^{-3}$ which can encode a spike kernel at distance exactly 1000 steps (Corollary 3.4). (*Right*) When $\Delta$ is set optimally, the proposed S4-FouT method is the best SSM as the theory predicts. When $\Delta$ is not set optimally, other methods perform better, including the simple diagonal methods proposed in (Gu et al., 2022b).

Figure 4: (**Delay Task.**) A synthetic memorization task: definition (Fig. 4a) and results (Fig. 4b).

## 5 DISCUSSION

This work improves the HiPPO framework, generalizing it to any set of orthonormal basis functions as projection operators. This led to a better understanding of existing models (clarification of the mechanisms underlying the original S4 model) as well as new variants (SSMs producing Fourier basis functions). In addition, we use our new framework to give principled explanations of other components such as the timescale and initialization, leading to improved empirical results.

The theoretical insights provided by this work have been used to improve and extend SSMs in several directions. We showed that S4 produces exponentially-decaying kernels according to precise formulas (Corollary C.6), and Li et al. (2022) designed alternative exponentially-decaying CNN kernels inspired by this property. Another line of work on diagonal approximations to S4 all use insights from our theory to simplify and improve S4. DSS (Gupta, 2022) introduced a particular diagonal approximation which was empirically effective, and S4D (Gu et al., 2022b) proved that it produced the same kernels asymptotically as S4 (Corollary C.6). S5 (Smith et al., 2022) extended this to multi-input multi-output (MIMO) SSMs and showed that our recommendations for initialization of $\boldsymbol{C}$ and $\Delta$ are important even in the MIMO setting. We believe that the insights in this work will be useful both to understand the original S4 model, and produce better and simpler state space models.

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

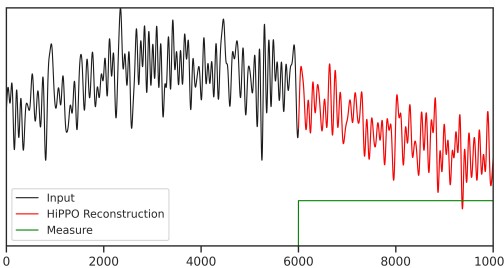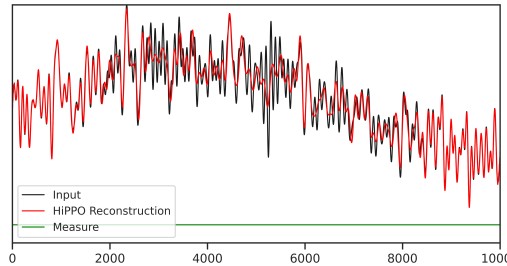

Figure 5: (**Prior HiPPO methods**) Given an input function $u(t)$ (black), HiPPO compresses it online into a state vector $x(t) \in \mathbb{R}^N$ via equation (1). Specific cases of HiPPO matrices $\boldsymbol{A},\boldsymbol{B}$ are derived so that at every time $t$, the history of $u$ up to time $t$ can be reconstructed linearly from $x(t)$ (red), according to a measure (green). (*Left*) The HiPPO-LegT method orthogonalizes onto the Legendre polynomials against a time-invariant uniform measure, i.e. sliding windows. (*Right*) The original HiPPO-LegS method is *not* time-invariant system. When used as a time-varying ODE $x' = \frac{1}{t}\boldsymbol{A}x + \frac{1}{t}\boldsymbol{B}u$, $x(t)$ represents the projection of the entire history of $u$ onto the Legendre polynomials. It was previously unknown how to interpret the time-invariant version of this ODE using the same $(\boldsymbol{A},\boldsymbol{B})$ matrices.

## A  RELATED WORK

We discuss in more detail the differences between this work and the previous results in this line of work.

### A.1  HiPPO OSSMs with Orthogonal Polynomial Kernels

HiPPO is an online function reconstruction framework theoretically motivated and described in (Gu et al., 2020) and expanded on in (Gu et al., 2021). By projecting sequence data onto polynomial bases, a function's history can be represented in a latent space.

Every measure $\mu$ (with some mild restrictions) in the finite[1] interval[2] $[-1,1]$ induces a unique sequences of orthogonal polynomials (OPs) $p_0(x), p_1, \dots$ satisfying $\deg(p_i) = i$ and

$$\langle p_i, p_j \rangle_\mu = \int_{-1}^{1} p_i(x)p_j(x)d\mu(x) = \delta_{ij} \qquad \text{for all } i, j, \tag{8}$$

where $\delta_{ij} = 1$ if $i = j$ and 0 otherwise.

This sequence forms an *OP family*. For a function $u$, HiPPO gives a compressed representation the history of $u$ in the interval $[t - \theta(t), t]$ in the $N$ coefficients given by $(0 \le n < N)$:

$$x_n(t) = \int_{t-\theta(t)}^{t} u(t)p_n(t,s)\mu(t,s)ds. \tag{9}$$

where $p_n(t,s)$ and $\mu(t,s)$ are transformations of the OP family onto the interval $[0,t]$. Specifically, one can choose $p_n(t,s) = p_n\left(\frac{2(s-t)}{\theta(t)} + 1\right)$ and $\mu(t,s) = \mu\left(\frac{2(s-t)}{\theta(t)} + 1\right)$. Note that Eq. (9) and Eq. (8) correspond exactly to Eq. (7).

When $\theta(t) = \theta$ for a fixed $\theta$, this corresponds to the 'truncated' window from (Gu et al., 2020) while the case of $\theta(t) = t$ considers the entire $[0,t]$ is the 'scaled' window case from (Gu et al., 2020). This is illustrated in Fig. 5.

(Gu et al., 2020) only considered the above for the case of $\mu(x) = 1$ for which the corresponding OP family is the Legendre polynomials. For $\theta(t) = \theta$ and $\theta(t) = t$ we get the LegT and LegS respectively in (Gu et al., 2020).

From the viewpoint of Definition 2, it is easy to see that choosing an OP family $p_i(x)$ and its measure $\mu$ defines an OSSM $(\mathbf{A}(t), \mathbf{B}(t))$. However, both (Gu et al., 2020) and (Gu et al.,

---

[1]The results on orthogonal polynomials also work for the infinite interval $[0,\infty)$ via the Laguerre polynomials but we ignore this case for simplicity but point out that (Gu et al., 2020) handles this case.

[2]In this work we presented everything for the finite interval $[0,1]$ but since $[-1,1]$ is more standard in the orthogonal polynomials and what was used in (Gu et al., 2020) we stick with $[-1,1]$ in this section. It is easy to move from one to another by an appropriate linear scaling of the argument.

2021) start from Eq. (9) and show that by differentiating Eq. (9) wrt $t$ one can derive the corresponding SSM:

$$x'(t) = \mathbf{A}(t)x(t) + \mathbf{B}(t)u(t).$$

Specifically, for the case of $\mu(x) = 1$ (i.e. Legendre) the above simplifies to:

1. For $\theta(t) = \theta$ (i.e. LegT) one gets

$$x'(t) = \frac{1}{\theta} \cdot \mathbf{A}x(t) + \frac{1}{\theta} \cdot \mathbf{B}u(t), \tag{10}$$

   where $\mathbf{A}$ and $\mathbf{B}$ are as in Eq. (5).

2. For $\theta(t) = t$ (i.e. LegS) one gets

$$x'(t) = \frac{1}{t} \cdot \mathbf{A}x(t) + \frac{1}{t} \cdot \mathbf{B}u(t), \tag{11}$$

   where $\mathbf{A}$ and $\mathbf{B}$ are as in Eq. (4).

## A.2 THE MYSTERY OF S4

The S4 paper (Gu et al., 2022b), used a weird 'mixture' of LegS and LegT. Specifically, in its experiments, it used the ODE in Eq. (10) but instead of using $\mathbf{A}$ and $\mathbf{B}$ are as in Eq. (5) it used $\mathbf{A}$ and $\mathbf{B}$ as in Eq. (4). However, LegS $\mathbf{A}$ and $\mathbf{B}$ from Eq. (4) had been derived for $\theta(t) = t$ instead of $\theta(t) = \theta$. In other words, **there was NO mathematical justification for the ODE used in S4**. One of our main results is to provide a solid mathematical justification for the ODE used in S4 (see Section 3.1).

## A.3 WHY OP KERNELS?

A crucial insight of the HiPPO framework is that the coefficients $x(t)$ is sufficient to recover $u$. This enables online predictions for end-to-end models. The intuition for this is basically that OPs form a complete basis for functions over $[-1,1]$. Specifically, given a $C^1$−smooth function $u : [-1,1] \rightarrow \mathbb{R}$ which is seen online, we wish to maintain a compressed representation of its history $u(s)_{\leq t} = u(s)_{s \leq t}$ at every time $t$.

For any infinite dimensional polynomial basis $p_0, p_1, ...$, we get the polynomial expansion of $u$:

$$u(t) = \sum_{n=0}^{\infty} x_n(t)p_n(t).$$

The truncated approximation of $u(t)$ at time $t = N$ is:

$$\hat{u}(t) = \sum_{n=0}^{N-1} x_n(t)p_n(t). \tag{12}$$

If $p_i$ is an OP family, then our approximation $\hat{u}(t)$ is guaranteed to be optimal. That is, as $N \rightarrow \infty$, $\hat{u}(t)$ becomes a perfect reconstruction of $u$ (i.e. the error with respect to the measure $\mu$ goes to 0 as $N \rightarrow \infty$). Further, given the measure $\mu(x)$ it is known that the OP family corresponding to $\mu$ gives the best possible approximation among all degree $N-1$ polynomial approximations.

The main insight of HiPPO (Gu et al., 2020) was to extend the framework above from the interval $[-1,1]$ to $[0,t]$ such that the approximation of Eq. (12) can be updated efficiently as $t$ increases.

In Appendix C.2, we expand the HiPPO framework to any set of differentiable orthogonal functions with respect to a given measure, and generalize the concepts behind LegS using the *time-warping* function $\sigma$. We use this to derive a general form for time-varying OSSMs, and give a mathematical interpretation of the LegS's state matrix.

## A.4 HIPPO VS LSSL

As discussed above, HiPPO can be thought of as a framework for deriving state space models corresponding to specific polynomial bases. The original paper (Gu et al., 2020) did not explicitly draw the connection to state space models, and also developed systems only

for a few particular cases which were called LegS (a time-varying system involving Legendre polynomials), LegT (a time-invariant system with the truncated Legendre polynomials), and LagT (involving Laguerre polynomials).

A follow-up paper on Linear State Space Layers (LSSL) (Gu et al., 2021) generalized these results to *all* orthogonal polynomial families, and also generalized the flexibility of the time-varying component. They produced SSMs $x'(t) = \boldsymbol{A}(t)x(t) + \boldsymbol{B}(t)u(t)$ where at all times $t$, $x(t)$ can be viewed as the projection of the history of $u(s)|_{s \leq t}$ onto orthogonal polynomials $p_n$ rescaled onto the interval $[t - \theta(t), t]$, where $\theta(t)$ is an arbitrary factor. (Indeed this is the form we outlined above.) This generalized all 3 cases of the original HiPPO paper.

### A.5    Legendre Memory Unit (Legendre Delay Network)

The HiPPO-LegT matrix (5) was first introduced as the LMU (Voelker, 2019; Voelker et al., 2019). The original motivation was to produce a state space model that approximates the **Delay Network**, which can be defined as the LTI system that transforms $u(t)$ into $u(t-1)$, i.e. lags the input by 1 time unit. This can also be defined as the system with impulse response $K(t) = \delta(t-1)$, i.e. it convolves by the convolutional kernel with a $\delta$ spike at time 1.

The connection between the Delay Network and Legendre polynomials was made in two steps. First, the transfer function of the ideal system is $\mathcal{L}[\delta(t-1)](s) = e^{-s}$ and must be approximated by a proper rational function to be represented as an SSM. Taking Padé approximants of this function yields "optimal" approximations by rational functions, which can then be distilled into a SSM $(\boldsymbol{A}, \boldsymbol{B}, \boldsymbol{C})$ whose transfer function $\boldsymbol{C}(s\boldsymbol{I} - \boldsymbol{A})^{-1}\boldsymbol{B}$ matches it. Second, the SSM basis $e^{t\boldsymbol{A}}\boldsymbol{B}$ for this system can be computed and found to match Legendre polynomials. However, despite making this connection and writing out formulas for this SSM, (Voelker, 2019) did not provide a complete proof of either of these two connections.

The preceding two steps that motivated the LDN can be informally written as the chain of transformations (i) transfer function $e^{-s} \rightarrow$ (ii) SSM $(\boldsymbol{A}, \boldsymbol{B}, \boldsymbol{C}) \rightarrow$ (iii) Legendre polynomials $e^{t\boldsymbol{A}}\boldsymbol{B}$. The HiPPO framework in a sense proceeded in the opposite direction. (Gu et al., 2020) started by defining the system that convolves with truncated Legendre polynomials, and with a particular differentiation technique showed that it could be written as a particular SSM which they called HiPPO-LegT. This SSM turned out to be the same (up to a minor change in scaling) as the original $(\boldsymbol{A}, \boldsymbol{B})$ defined by the LMU, thus proving the second of the two steps relating this particular SSM to the Legendre polynomials.

In this work, we show the final piece in this reverse chain of equivalences. In particular, we start from the LegT SSM $(\boldsymbol{A}, \boldsymbol{B}, \boldsymbol{C})$ and directly prove that its transfer function produces Padé approximants of the exponential. Our proof introduces new techniques in an inductive argument that can be applied to HiPPO SSMs beyond the LegT case, and relates them to continued fraction expansions of the exponential.

We comment on a minor difference between the parameterization of HiPPO-LegT and the LMU. The LMU is originally defined as

$$x'(t) = \frac{1}{\theta}\boldsymbol{A}x(t) + \frac{1}{\theta}\boldsymbol{B}u(t)$$

where $\theta$ is a hyperparameter that controls the length of the window. However, we point out that such constant scaling of the SSM is also controlled by the step size $\Delta$ as discussed in Section 3.3. Therefore $\theta$ is redundant with $\Delta$, so the LegT matrices defined in (Gu et al., 2020) and in this work do not have a concept of $\theta$. Additionally, in this work we redefine the LegT matrices $(\boldsymbol{A}, \boldsymbol{B})$ to be scaled by a factor of 2 to make them properly timescale normalized, using the theory developed in Section 3.3.

### A.6    Our framework

Compared to these works, our framework (Definition 2) simplifies and generalizes the concepts directly in terms of (time-varying) state space models. We define a more natural concept of **orthogonal SSM**, derive very general instantiations of it (Section 3.1), and flesh out its properties (Section 3.3). Our general result subsumes all prior cases including all cases of the LSSL as a direct corollary. Some concrete advantages include:

- It allows more flexible transformations of polynomial bases, such as including a change-of-basis inside the polynomials. The previously expained case of LegS is

an instance of this, which has basis functions $L(e^{-t})$ with an exponential change of basis, instead of vanilla polynomials.

- It can be applied to non-polynomial bases, such as the truncated Fourier basis FouT.
- It does not require considering multiple cases depending on where the basis functions are supported. Instead, we handle this by considering discontinuities in the basis functions.

### A.7 APPLICATION IN DEEP LEARNING SYSTEMS

While the preceding discussion covers theoretical interpretations of SSMs, S4 (Gu et al., 2022a) (and its predecessor LSSL (Gu et al., 2021)) are the application of these SSMs to deep learning. In comparison to prior works such as the LMU and HiPPO which require a pre-determined system $(\boldsymbol{A},\boldsymbol{B})$ and incorporate them naively into an RNN, LSSL and S4 use a full state space model $(\boldsymbol{A},\boldsymbol{B},\boldsymbol{C})$ as a completely trainable deep learning layer. Doing this required resolving computational problems with the SSM, which was the main focus of S4. Specifically, the results in HiPPO (Gu et al., 2020) and LSSL (Gu et al., 2021) only guaranteed theoretical efficiency: in particular, they showed how the various computations can be done with near-linear number of *arithmetic operations* and does not specifically guarantee any sort of *numerical stability*—the main theoretical contribution of S4 (Gu et al., 2022a) was to give a numerically stable algorithm.

In this work, we make a distinction between HiPPO, which is the theoretical derivation and interpretation of particular SSMs $(\boldsymbol{A},\boldsymbol{B})$, and S4, which is the incorporation of those SSMs as a trainable deep learning layer with a particular algorithm.

## B EXPERIMENT DETAILS AND ADDITIONAL EXPERIMENTS

### B.1 SYNTHETIC RECONSTRUCTION TASK

We construct a synthetic **Reconstruction Task** against a uniform measure. The input is a white noise sequence $u \in \mathbb{R}^{4000}$. We use a single layer linear S4 model with state size $N = 256$ and $H = 256$ hidden units. Models are required to use their output at the last time step, a vector $y_{4000} \in \mathbb{R}^{256}$, to reconstruct the last 1000 elements of the input with a linear probe. Concretely, the loss function is to minimize $\|u_{3000:4000} - \boldsymbol{W} y_{4000}\|_2^2$, where $\boldsymbol{W} \in \mathbb{R}^{1000 \times 256}$ is a learned matrix. Models are trained with the Adam optimizer with learning rate 0.001 for 20 epochs.

Fig. 6 shows that S4-LegT and S4-FouT, the methods that theoretically reconstruct against a uniform measure, are far better than other methods. We include the new diagonal variants (S4D) proposed in (Gu et al., 2022b), which are simpler SSM methods that generally perform well but do not learn the right function on this task. We also include a method **S4-(LegS/FouT)** which combines both LegS and FouT measures by simply initializing half of the SSM kernels to each. Despite having fewer S4-FouT kernels, this still performs as well as the pure S4-FouT initializing.

### B.2 DELAY (CONTINUOUS COPYING) TASK

The Delay Task consists of input-output pairs where the input is a white noise signal of length 4000 bandlimited to 1000 Hz. The output is the same signal shifted by 1000 steps (Fig. 4a). We use single layer linear SSMs with $H = 4$ hidden units and state size $N = 1024$. Models are trained with the Adam optimizer with learning rate 0.001 for 20 epochs.

### B.3 LONG RANGE ARENA

The settings for LRA use the same hyperparameters in (Gu et al., 2022b). A more detailed protocol can be found in (Gu et al., 2022b). To be self-contained, we recreate the same table of parameters in Table 2.

## C PROOF DETAILS

We furnish the missing proofs from Section 2 in Appendix C.1. We will describe our general framework and results in Appendix C.2, and prove the results in Sections 3.1 to 3.3 in Appendices C.3 to C.5 respectively.

| | (Δ min, Δ max) | | (Δ min, Δ max) | |
|---|---|---|---|---|
| | (1e-3, 2e-3) | (2e-3, 2e-3) | (1e-3, 1e-1) | (2e-3, 1e-1) |
| **S4-LegS** | -6.2581 | -6.6328 | -3.5505 | -3.3017 |
| **S4-LegT** | -7.4987 | -8.1056 | -2.9729 | -2.6557 |
| **S4-FouT** | -7.4889 | -8.3296 | -3.0062 | -2.6976 |
| **S4-(LegS/FouT)** | -7.4992 | -8.3162 | -3.4784 | -3.2628 |
| **S4D-LegS** | -6.1528 | -6.184 | -3.8773 | -3.6317 |
| **S4D-Inv** | -5.9362 | -6.0986 | -4.1402 | -3.7912 |
| **S4D-Lin** | -7.1233 | -6.6483 | -3.973 | -3.5991 |
| **S4D-(Inv/Lin)** | -6.839 | -6.705 | -4.325 | -3.8389 |

Figure 6: Log-MSE after training on the Reconstruction Task. (*Left*) When the timescales $\Delta$ are set appropriately for this task, the methods that theoretically reconstruct against a uniform measure (LegT and FouT) are much better than alternatives, achieving MSE more orders of magnitude lower than other SSM initializations. (*Right*) Interestingly, when the timescales $\Delta$ are not set correctly, these methods (LegT and FouT) actually perform worst and the diagonal methods introduced in (Gu et al., 2022b) perform best.

Table 2: The values of the best hyperparameters found for LRA. LR is learning rate and WD is weight decay. BN and LN refer to Batch Normalization and Layer Normalization.

| | Depth | Features $H$ | Norm | Pre-norm | Dropout | LR | Batch Size | Epochs | WD | $(\Delta_{min}, \Delta_{max})$ |
|---|---|---|---|---|---|---|---|---|---|---|
| **ListOps** | 8 | 128 | BN | False | 0 | 0.01 | 50 | 40 | 0.05 | (0.001,0.1) |
| **Text** | 6 | 256 | BN | True | 0 | 0.01 | 16 | 32 | 0.05 | (0.001,0.1) |
| **Retrieval** | 6 | 256 | BN | True | 0 | 0.01 | 64 | 20 | 0.05 | (0.001,0.1) |
| **Image** | 6 | 512 | LN | False | 0.1 | 0.01 | 50 | 200 | 0.05 | (0.001,0.1) |
| **Pathfinder** | 6 | 256 | BN | True | 0 | 0.004 | 64 | 200 | 0.03 | (0.001,0.1) |
| **Path-X** | 6 | 256 | BN | True | 0 | 0.0005 | 32 | 50 | 0.05 | (0.0001,0.01) |

### C.1 PROOFS FROM BACKGROUND

This corresponds to results from Section 2.

Recall that for OSSMs, $(p,\omega)$ and $K$ are uniquely determined by each other, so we can refer to an OSSM by either. One direction is obvious: $(p,\omega)$ determine $K$ via $K_n(t,s) = p_n(t,s)\omega(t,s)$.

**Proposition 6.** *If a set of kernel functions satisfies $K_n(t,s) = p_n(t,s)\omega(t,s)$ where the functions $p_n$ are complete and orthogonal w.r.t. $\omega$ (equation (7) right), $p$ and $\omega$ are unique.*

*Proof of Proposition 6.* Suppose for the sake of contradiction that there is a second basis and measure $q_n, \mu$ such that $q_n$ is complete and orthogonal w.r.t. $\mu$, and $K_n = q_n\mu$. By completeness, there are coefficients $c_{\ell,k}$ such that

$$p_\ell = \sum_k c_{\ell,k} q_k.$$

Then

$$\int p_\ell q_j \mu = \int \sum_k c_{\ell,k} q_k q_j \mu = \sum_k c_{\ell,k} \delta_{kj} = c_{\ell,j}.$$

But $q_j\mu = K_j = p_j\omega$, so

$$\int p_\ell q_j \mu = \int p_\ell p_j \omega = \delta_{\ell j}.$$

So $c_{\ell,j} = \delta_{\ell,j}$ which implies that $p_\ell = q_\ell$ for all $\ell$, as desired. □

The main barrier to using Proposition 1 for function reconstruction is that SSMs are in general not OSSMs. For example, even though we will show that (4) is an TOSSM, and that unitary conjugation of a TOSSM is a TOSSM (Section 3.3), its diagonal matrix of eigenvalues is not a TOSSM. This both shows the existence of an SSM that is not an OSSM, and also implies that general conjugation does not preserve TOSSMs.

**Proposition 7.** *There is no TOSSM with the diagonal state matrix $\boldsymbol{A} = \text{diag}\{-1,-2,...\}$.*

*Proof of Proposition 7.* The SSM kernels are $K_n(t) = e^{-t(n+1)} \boldsymbol{B}_n$. Assume $\boldsymbol{B}_n \neq 0$ so that the kernels are not degenerate.

Suppose for the sake of contradiction that this was a TOSSM with measure $\omega(t)$. Then we must have

$$\int K_n(s) K_m(s) \omega(t)^{-1} ds = \delta_{n,m}$$

Plugging in $n=1, m=1$ and $n=0, m=2$ gives

$$1 = \int e^{-2t} \boldsymbol{B}_1 e^{-2t} \boldsymbol{B}_1 \omega(t)^{-1} ds = \boldsymbol{B}_1 \boldsymbol{B}_1 \int e^{-4t} \omega(t)^{-1} ds$$

$$0 = \int e^{-1t} \boldsymbol{B}_0 e^{-3t} \boldsymbol{B}_2 \omega(t)^{-1} ds = \boldsymbol{B}_0 \boldsymbol{B}_2 \int e^{-4t} \omega(t)^{-1} ds$$

This is clearly a contradiction. $\qquad\square$

## C.2 GENERAL THEORY

Consider a measure supported on $[0,1]$ with density $\omega(t)\mathbb{I}(t)$, where $\mathbb{I}(t)$ is the indicator function for membership in the interval $[0,1]$. Let the measure be equipped with a set of orthonormal basis functions $p_0, ..., p_{N-1}$, i.e.

$$\int p_j(s) p_k(s) \omega(s) \mathbb{I}(s) ds = \delta_{jk}, \tag{13}$$

where the integrals in this paper are over the range $[-\infty, \infty]$, unless stated otherwise.

This is sufficient to derive an OSSM based on the HiPPO technique. The generalized HiPPO framework demonstrates how to build (T)OSSMs utilizing *time warping* to shape the time interval and *tilting* to construct new sets of orthogonal basis functions.

Given an general interval $[\ell, r]$, we will use the notation $\mathbb{I}[\ell, r]$ to denote the indicator function for the interval $[\ell, r]$– we will drop the interval if $\ell = 0, r = 1$.

We will need the notion of a *"time warping"* function $\overline{\sigma}$ as follows:

**Definition 5.** *A* time warping function *is defined as*

$$\overline{\sigma}(t,s) : (-\infty, t] \to [0,1]$$

*such that $\overline{\sigma}(t,t) = 1$.*

*We will be using a special case of time-warping function, which we say* has a discontinuity at $t_0$ *for some $t_0 \in (-\infty, t]$:*

$$\overline{\sigma}(t,s) = \mathbb{I}[t_0, t] \sigma(t,s), \tag{14}$$

*such that*

$$\frac{\partial}{\partial t} \left( \frac{\partial}{\partial s} \sigma_s(t,s) \right) = c(t) \frac{\partial}{\partial s} \sigma(t,s). \tag{15}$$

*We allow for $t_0 = -\infty$, in which case we think of the interval $[t_0, t]$ as $(-\infty, t]$.*

Before proceeding, let us clarify our notation. We will use $\sigma_t$ and $\sigma_s$ to denote the partial derivatives $\frac{\partial}{\partial t}\sigma(t,s)$ and $\frac{\partial}{\partial s}\sigma(t,s)$ respectively. We will drop the parameters $(t,s)$ and use $f$ instead of $f(t,s)$ when it is clear from context to reduce notational clutter. Further, we will extend this notation to function composition, i.e. write $g \circ f(t,s)$ as $g(f)$ and function product, i.e. use $fgh$ instead of $f(t,s)g(t,s)g(t,s)$. Finally, we'll shorten $fgh \circ \phi(t,s)$ as $fgh(\phi)$.

We also define the tilting $\chi$ and show that regardless of warping, we can construct a new orthogonal basis (note that the result holds for warping functions as in (14) and not just those as in (15)).

**Lemma C.1.** *For the set of orthonormal functions $\{p_n\}_{n=0}^{N-1}$ orthogonal over measure $\omega I$, the set of basis functions*

$$q_k^t(\sigma(t,s)) = \chi(t,s) p_k(\sigma(t,s))$$

*are orthogonal over the measure*

$$\mu(t,s) = \omega(\sigma(t,s)) \mathbb{I}[t_0, t](s) \sigma_s(t,s) \chi(t,s)^{-2}$$

*for time-warping function $\sigma$ satisfying (14) and any $\chi(t,s)$ that is non-zero in its support.*

*Proof.* Consider the following sequence of equalities:

$$\int p_j(\sigma)p_k(\sigma)\omega(\sigma)\mathbb{I}[t_0,t]\sigma_s ds = \int_{t_0}^{t} p_j(\sigma)p_k(\sigma)\omega(\sigma)\sigma_s ds$$

$$= \int_{\sigma(t,t_0)}^{\sigma(t,t)} p_j(y)p_k(y)\omega(y)dy$$

$$= \int_{\sigma(t,t_0)}^{\sigma(t,t)} p_j(y)p_k(y)\omega(y)dy$$

$$= \int_{0}^{1} p_j(y)p_k(y)\omega(y)dy$$

$$= \int p_j(y)p_k(y)\omega(y)\mathbb{I}(y)dy$$

$$= \delta_{jk}.$$

In the above, the second equality follows from the substitution $y \leftarrow \sigma(t,s)$ and hence $dy = \sigma_s ds$ and the final equality follows from (13). Then since $\chi(t,s)$ is always non-zero, we have

$$\int (\chi p_j(\sigma))(\chi p_k(\sigma))\omega(\sigma)\mathbb{I}[t_0,t]\sigma_s\chi^{-2}ds = \delta_{jk},$$

as desired. $\qquad\square$

Without loss of generality, we can split $\chi$ into a product

$$\chi(t,s) = \frac{1}{\psi(\sigma(t,s))\phi(t,s)} \tag{16}$$

of one part that depends on $\sigma$ and another arbitrary component.

**Time Warped HiPPO.** Since we have an orthonormal basis and measure, we can try to derive the (T)OSSM. For a given input signal $u(t)$, the HiPPO coefficients are defined as the projections.

$$x_n(t) = \langle u, \chi p_n \rangle_\mu$$

$$= \int u(s)\cdot\chi\cdot(p_n\omega)(\sigma)\mathbb{I}[t_0,t]\sigma_s\chi^{-2}ds$$

defined as inner product of $u(t)$ with the tilted basis functions $\chi p_n$ with respect to the measure $\mu$ as defined in Lemma C.1. For additional convenience, we use the decomposition $\chi = \psi^{-1}\phi^{-1}$ from (16) to get:

$$x_n(t) = \int u(s)\cdot(p_n\omega\psi)(\sigma)\mathbb{I}[t_0,t]\sigma_s\phi ds. \tag{17}$$

The HiPPO technique is to differentiate through this integral in a way such that it can be related back to $x_n(t)$ and other $x_k(t)$. We require for every $n$, we require that there are a set of coefficients $\{\gamma_{nk}\}_{k=0}^{N-1}$ such that

$$\sigma_t(p_n\omega\psi)'(\sigma) = \sum_{k=0}^{N-1} \gamma_{nk}(p_n\omega\psi)(\sigma) \tag{18}$$

and for tilting component $\phi$

$$\frac{d}{dt}\phi(t,s) = d(t)\phi(t,s). \tag{19}$$

**Theorem 8.** *Consider a set of basis functions $p_n$ orthogonal over $\omega$, time warping $\overline{\sigma}(t,s)$ as in (14), (15), and tilting $\chi$ as in (16) and (19) with the functions $\sigma, p_n, \omega, \psi$ obeying (18). If*

$\frac{dt_0}{dt} \neq 0$, *further assume that for some vector* $\boldsymbol{A'}$, *we have as* $N \to \infty$,

$$u(t_0) = \overline{c} \sum_{k=0}^{N-1} \boldsymbol{A'}_k \cdot x_k(t) + \overline{d} u(t). \tag{20}$$

*Then* $(\boldsymbol{A}^0 + (c(t) + d(t))\boldsymbol{I} - \overline{c}\boldsymbol{D}(\boldsymbol{A'})^\top, \boldsymbol{B} - \overline{d}\boldsymbol{D})$ *is an OSSM for basis functions* $\chi p_n(\sigma)$ *with measure* $\omega \mathbb{I}[t_0, t]\sigma_s \chi^{-2}$ *where*

$$\boldsymbol{A}^0_{nk} = \gamma_{nk}$$

*with* $\gamma_{nk}$ *as in* (18),

$$\boldsymbol{D}_n = (p_n \omega \psi)(\sigma(t, t_0))(\sigma_s \phi)(t, t_0) \cdot \frac{dt_0}{dt},$$

*and*

$$\boldsymbol{B}_n = (p_n \omega \psi)(1)(\sigma_s \phi)(t, t).$$

*Proof.* Applying the Leibniz rule to (17), we get
$$x'_n(t) = x_n^{(0)}(t) + x_n^{(1)}(t) + x_n^{(2)}(t) + x_n^{(3)}(t),$$
where

$$x_n^{(0)}(t) = \int u(s) \cdot \sigma_t(p_n \omega \psi)'(\sigma)\mathbb{I}[t_0, t]\sigma_s \phi ds$$

$$x_n^{(1)}(t) = \int u(s) \cdot (p_n \omega \psi)(\sigma)\mathbb{I}[t_0, t]\left[\frac{\partial}{\partial t}(\sigma_s \phi)\right] ds$$

and the $x_n^{(2)}(t) + x_n^{(3)}(t)$ terms capture the term we get when differentiating $\mathbb{I}[t_0, t]$.

Let us consider each term separately. The first term

$$x_n^{(0)}(t) = \int u(s) \cdot \sigma_t(p_n \omega \psi)'(\sigma)\mathbb{I}[t_0, t]\sigma_s \phi ds \tag{21}$$

corresponds to the differentiation of the basis functions and measure. In order to relate this to $\{x_k(t)\}$, it suffices that $\sigma_t(p_n \omega \psi)'(\sigma)$ satisfies (18) which implies that when we vectorize this, we get $x^{(0)}(t) = \boldsymbol{A}^0 \cdot x(t)$.

For additional warping and tilting terms, we consider

$$x_n^{(1)}(t) = \int u(s) \cdot (p_n \omega \psi)(\sigma)\mathbb{I}[t_0, t]\left[\frac{\partial}{\partial t}(\sigma_s \phi)\right] ds.$$

To reduce this term to $x_n(t)$, recall from (15) that

$$\partial_t(\sigma_s) = c(t)\sigma_s.$$

Then the above and (19) imply
$$\partial_t(\sigma_s \phi) = c(t)(\sigma_s \phi) + d(t)(\sigma_s \phi)$$

where $c(t), d(t)$ are defined as in (15) and (19).

We will end up with $x_n^{(1)}(t) = (c(t) + d(t))x_n(t)$. This leads to the the vectorized form $x^{(1)}(t) = (c(t) + d(t))\boldsymbol{I}x(t)$.

We now need to handle

$$x_n^{(2)}(t) + x_n^{(3)}(t) = \int u(s) \cdot (p_n \omega \psi)(\sigma)\left[\frac{\partial}{\partial t}\mathbb{I}[t_0, t]\right](\sigma_s \phi) ds. \tag{22}$$

For the above note that
$$\mathbb{I}[t_0, t](s) = H(s - t_0) - H(s - t),$$
where $H(x)$ is the "heaviside step function." It is know that $H'(x) = \delta(x)$, which implies

$$\frac{\partial}{\partial t}\mathbb{I}[t_0, t] = \delta(s - t) - \frac{dt_0}{dt}\delta(s - t_0).$$

Using the above in RHS of (22), we separate out $x_n^{(2)}(t)$ and $x_n^{(3)}(t)$ as follows. First, define

$$x_n^{(2)}(t) = \int u(s) \cdot (p_n \omega \psi)(\sigma) \delta(s-t) \sigma_s \phi \, ds$$
$$= u(t) \cdot (p_n \omega \psi)(\sigma(t,t))(\sigma_s \phi)(t,t)$$
$$= u(t) \cdot (p_n \omega \psi)(1)(\sigma_s \phi)(t,t).$$

In the last equality, we have used the fact that $\sigma(t,t) = \overline{\sigma}(t,1) = 1$ by definition. It follows that in vectorized form we have $x^{(2)}(t) = \boldsymbol{B}u(t)$.

Finally, define

$$x_n^{(3)}(t) = -\int u(s) \cdot (p_n \omega \psi)(\sigma) \delta(s-t_0) \frac{dt_0}{dt} \sigma_s \phi \, ds$$
$$= -u(t_0) \cdot (p_n \omega \psi)(\sigma(t,t_0))(\sigma_s \phi)(t,t_0) \cdot \frac{dt_0}{dt}$$

If $\frac{dt_0}{dt} = 0$, then we have $\boldsymbol{D} = \boldsymbol{0}$ and hence we have $x^{(3)}(t) = \boldsymbol{0} = -\overline{c}\boldsymbol{D}(\boldsymbol{A}')^\top x(t) - \overline{d}\boldsymbol{D}u(t)$

If $\frac{dt_0}{dt} \neq 0$, then as $N \to \infty$, from (20), the above comes out to

$$x_n^{(3)}(t) = -\left( \overline{c} \sum_{k=0}^{N-1} \boldsymbol{A}_k' \cdot x_k(t) + \overline{d}u(t) \right) \cdot (p_n \omega \psi)(\sigma(t,t_0))(\sigma_s \phi)(t,t_0) \cdot \frac{dt_0}{dt}$$

It follows that in vectorized form we have $x^{(3)}(t) = -\overline{c}\boldsymbol{D}(\boldsymbol{A}')^\top x(t) - \overline{d}\boldsymbol{D}u(t)$. The result follows after combining the terms.

$\square$

We see that the behavior of is the model is dictated by $t_0$. In particular, in this paper, we will consider two special cases.

**Corollary C.2** ($t_0$ independent of $t$). *The SSM* $((\boldsymbol{A} + c(t) + d(t)\boldsymbol{I}), \boldsymbol{B})$ *satisfying conditions of Theorem 8 with $t_0$ independent of $t$, is an OSSM for basis functions $\chi p_n(\sigma)$ with measure $\omega \mathbb{I}[t_0,t]\sigma_s\chi^{-2}$ where $\boldsymbol{A} = \gamma_{nk}$ as in (18) and $\boldsymbol{B}_n = (p_n\omega\psi)(1)(\sigma_s\phi)(t,t)$.*

*Proof.* Follows from Theorem 8. Since $t_0$ is independent of $t$, then $\frac{dt_0}{dt} = 0$, and $\boldsymbol{D} = \boldsymbol{0}$. $\square$

**Corollary C.3** ($t_0 = t - \theta$). *The SSM* $(\boldsymbol{A}^0 + (c(t) + d(t))\boldsymbol{I} - \overline{c}\boldsymbol{D}\boldsymbol{A}', \boldsymbol{B} - \overline{d}\boldsymbol{D})$ *satisfying conditions of Theorem 8 with $t_0 = t - \theta$ for a fixed $\theta$, is an OSSM with basis functions $\chi p_n(\sigma)$ with measure $\omega \mathbb{I}[t_0,t]\sigma_s\chi^{-2}$ where $\boldsymbol{A}_{nk}^0 = \gamma_{nk}$ as in (18), $\boldsymbol{D}_n = (p_n\omega\psi)(\sigma(t,t-\theta))(\sigma_s\phi)(t,t-\theta)$, and $\boldsymbol{B}_n = (p_n\omega\psi)(1)(\sigma_s\phi(t,t))$.*

*Proof.* This follows directly from Theorem 8 by setting $t_0 = t - \theta$. $\square$

## C.3 Time-Varying Windows

### C.3.1 Explanation of S4-Legs

Consider the case when

$$\sigma = \omega^{-1},$$

i.e. the measure is completely "tilted" away, and let

$$\frac{\partial}{\partial t}\sigma(t,s) = a(t)\sigma(t,s) + b(t). \tag{23}$$

Let's consider the special case of (23) where $b(t) = 0$. This is most generally satisfied by

$$\sigma(t,s) = \exp(a(t) + z(s)).$$

Note that the condition $\sigma(t,t) = 1$ forces $z = -a$. Hence, we have

$$\sigma(t,s) = \exp(a(s) - a(t)). \tag{24}$$

We now consider the following special case of Corollary C.2:

**Corollary C.4.** *Let $\eta \geq 0$. The SSM $(-a'(t)(\boldsymbol{A}+(\eta+1)\boldsymbol{I}), a'(t)\boldsymbol{B})$, where $t_0$ is independent of $t$, is an OSSM for basis functions and measure*

$$\frac{\omega(\sigma)}{\sigma^\eta}p_n(\sigma) \qquad \omega(t,s) = \mathbb{I}(\sigma)\frac{a'(s)\sigma^{2\eta+1}}{\omega(\sigma)}$$

*where $\sigma$ satisfies* (24)*,*

$$\phi(t,s) = \exp(\eta a(s) - \eta a(t)) = \sigma^\eta, \tag{25}$$

$\boldsymbol{A} = \alpha_{nk}$ *such that*

$$yp'_n(y) = \sum_{k=0}^{n-1}\alpha_{nk}p_k(y) \tag{26}$$

*and*

$$\boldsymbol{B}_n = p_n(1).$$

*Proof.* Given a orthonormal basis $p_0, p_1, ..., p_{N-1}$ with respect to a measure $\omega$. Note that time-warping function $\sigma$ satisfying (24) implies that $\sigma_s = a'(s)\sigma$.

We fix tilting $\chi(t,s) = \frac{\omega(\sigma)}{\sigma^\eta}$, which in turn follows by setting

$$\psi = \omega^{-1}.$$

We show shortly that we satisfy the pre-conditions of Corollary C.2, which implies (with our choice of $\chi$ and $\sigma$) that we have an OSSM with basis functions $p_n(t,s) = \frac{\omega(\sigma)}{\sigma^\eta}p_n(\sigma)$ and measure

$$\omega(t,s) = \omega(\sigma(t,s))\mathbb{I}[t_0,t]\sigma_s(t,s)\chi(t,s)^{-2}$$
$$= \mathbb{I}[t_0,t]\frac{a'(s)\sigma^{2\eta+1}}{\omega(\sigma)}$$

To complete the proof, we show that out choice of parameters above satisfies the conditions of Corollary C.2 (by showing they satisfy the conditions of Theorem 8). We verify that $\sigma$ and $\phi$ satisfy (15) and (19), noting that

$$\partial_t(\sigma_s) = -a'(t)\sigma_s,$$

and

$$\partial_t(\phi) = -\eta a'(t)\phi.$$

This implies that setting $c(t) = -a'(t)$ and $d(t) = -\eta a'(t)$ is enough to satisfy (15) and (19).

Further, note that (24) and the fact that $\psi = \omega^{-1}$ imply that

$$\sigma_t(p_n\omega\psi)'(\sigma) = -a'(t)\sigma p'_n(\sigma).$$

It follows that (18) is satisfied as long as

$$\sigma p'_n(\sigma) = \sum_{k=0}^{n-1}\alpha_{nk}p_k(\sigma)$$

for some set of coefficients $\{\alpha_{nk}\}_{k=0}^{N-1}$, which is exactly (26). This implies the $\gamma_{nk}$ in Corollary C.2 satisfy.

$$\gamma_{nk} = -a'(t)\alpha_{nk}.$$

Let $\boldsymbol{A}$ be the matrix such that $\boldsymbol{A}_{nk} = -\alpha_{nk}$ and then note that $-a'(t)(\boldsymbol{A}+(\eta+1)\boldsymbol{I})$ is exactly the first parameter of the SSM in Corollary C.2. Similarly, recall in Corollary C.2

$$\boldsymbol{B}_n = p_n(1)(\sigma_s\phi)(t,t)$$
$$= p_n(1)a'(t),$$

where the final equality follows since in our case, $\sigma_s(t,t) = a'(t)\exp(a(t) - a(t)) = a'(t)$. Overloading notation and letting $\boldsymbol{B}_n = p_n(1)$, all conditions of Corollary C.2 hold, from which the claimed result follows. $\square$

We are particularly interested in the following two special cases of Corollary C.4.

**Corollary C.5.** *The SSM $(-\frac{1}{t}(\boldsymbol{A}+\boldsymbol{I}),\frac{1}{t}\boldsymbol{B})$ is a OSSM for basis functions $p_n(\frac{s}{t})\omega(\frac{s}{t})$ with measure $\frac{1}{t}\mathbb{I}[t_0,t]\left(\frac{s}{t}\right)\cdot\omega(\frac{s}{t})$ where $\boldsymbol{A}=\alpha_{nk}$ as in (26) and $\boldsymbol{B}_n=p_n(1)$.*

*Proof.* Letting $a'(t)=\frac{1}{t}$ implies that $a(t)=\ln t$. Then we can observe that is a case of Corollary C.4 with time warping

$$
\begin{aligned}
\sigma(t,s) &= \exp(-\ln t + \ln s) \\
&= \exp(\ln(s/t)) \\
&= \frac{s}{t}.
\end{aligned}
$$

We set $\eta=0$ in Corollary C.4, which in turn sets $\phi=\sigma^0=1$. This gives the tilting

$$
\begin{aligned}
\chi &= \phi^{-1}\psi^{-1} \\
&= \omega.
\end{aligned}
$$

Then by Corollary C.4, it follows that that we can use $\sigma$ and $\chi$ to build an OSSM with basis functions

$$
\frac{\omega(\sigma)}{\sigma^\eta}p_n(\sigma)=\omega(\frac{s}{t})\cdot p_n(\frac{s}{t})
$$

with measure

$$
\mathbb{I}(\sigma)\frac{a'(s)\sigma^{2\eta+1}}{\omega(\sigma)}=\frac{1}{t}\mathbb{I}(\sigma)\frac{\sigma}{\omega(\sigma)}.
$$

Then the result follows. $\square$

**Corollary C.6.** *The SSM $(-(\boldsymbol{A}+\boldsymbol{I}),\boldsymbol{B})$ is a OSSM for basis functions $p_n(e^{s-t})\omega(e^{s-t})$ with measure $\omega=\mathbb{I}[t_0,t](e^{s-t})\frac{e^{s-t}}{\omega(e^{s-t})}$ where $\boldsymbol{A}=\alpha_{nk}$ as in (26) and $\boldsymbol{B}_n=p_n(1)$.*

*Proof.* This is a case of Corollary C.4 where $a'(t)=1$, $\sigma=\exp(s-t)$, and we pick $\eta=0$, implying that $\phi=\sigma^0=1$. It follows that

$$
\begin{aligned}
\chi &= \phi^{-1}\psi^{-1} \\
&= \omega.
\end{aligned}
$$

Utilizing Corollary C.4, we can use $\sigma$ and $\chi$ to build an OSSM with basis functions

$$
\frac{\omega(\sigma)}{\sigma^\eta}p_n(\sigma)=\omega(\exp(s-t))\cdot p_n(\exp(s-t))
$$

with measure

$$
\mathbb{I}(\sigma)\frac{a'(s)\sigma^{2\eta+1}}{\omega(\sigma)}=\mathbb{I}(\sigma)\frac{\exp(s-t)}{\omega(\exp(s-t))}.
$$

This gives us our final result. $\square$

Next we instantiate Corollary C.4 to prove Corollary 3.1. (Even though strictly not needed, we instantiate Corollary C.6 and Corollary C.5 to prove Theorem 3 and Corollary 3.3.) To that end, we will need the following result:

**Lemma C.7.** *Let the Legendre polynomials orthonormal over the interval $[0,1]$ be denoted as $L_n$. Then*

$$
yL_n'(y)=nL_n(y)+\sqrt{2n+1}\left(\sum_{k=0}^{n-1}\sqrt{2k+1}L_k(y)\right), \tag{27}
$$

$$L'_n(y) = 2\sqrt{2n+1}\left(\sum_{0 \le k \le n-1, n-k \text{ is odd}} \sqrt{2k+1}L_k(y)\right), \tag{28}$$

*and*

$$L_n(0) = (2n+1)^{\frac{1}{2}}(-1)^n \text{ and } L_n(1) = (2n+1)^{\frac{1}{2}}. \tag{29}$$

*Proof.* The Legendre polynomials satisfy the following orthogonality condition over $[-1,1]$:

$$\int_{-1}^{1} P_m(z)P_n(z)dz = \frac{2}{2n+1}\delta_{mn}.$$

Let us denote the normalized Legendre polynomials orthogonal over $[-1,1]$ as $\lambda_n P_n(z)$ where $\lambda_n = \sqrt{\frac{2n+1}{2}}$. To orthogonalize them over $[0,1]$, let $y = \frac{1+z}{2}$. It follows that $z = 2y-1$, $dz = 2dy$. Note that we then have

$$\int_{-1}^{1} P_m(z)P_n(z)dz = \int_{0}^{1} 2P_m(2y-1)P_n(2y-1)dy.$$

This implies that

$$\int_{-1}^{1} \frac{2n+1}{2} \cdot 2P_m(2y-1)P_n(2y-1)dy = \delta_{mn}.$$

Then if we let

$$L_n(y) = \sqrt{2}\lambda_n P_n(2y-1) = \sqrt{2n+1}P_n(2y-1), \tag{30}$$

then we have an a set of functions over $[0,1]$ such that

$$\int_{0}^{1} L_m(y)L_n(y)dy = \delta_{mn}.$$

From (Chihara, 2011, (2.8), (2.9)), note that $P_n(-1) = (-1)^n$ and $P_n(1) = 1$. This implies that

$$L_n(0) = \sqrt{2n+1}P_n(-1), \quad L_n(1) = \sqrt{2n+1}P_n(1).$$

Finally note that (30) implies:

$$\begin{aligned}L'_n(y) &= 2\sqrt{2n+1}P'_n(2y-1) \\ &= 2\sqrt{2n+1}P'_n(z).\end{aligned}$$

From (Gu et al., 2020, 7), we get

$$P'_n(z) = \sum_{0 \le k \le n-1, n-k \text{ is odd}} (2k+1)P_k(z).$$

Using (30) on the above, we get (28).

We now consider

$$\begin{aligned}yL'_n(y) &= 2y\sqrt{2n+1}P'_n(z) \\ &= (1+z)\sqrt{2n+1}P'_n(z).\end{aligned}$$

From (Gu et al., 2020, 8), we get

$$(z+1)P'_n(z) = nP_n(z) + \sum_{k=0}^{n-1}(2k+1)P_k(z).$$

Then the above becomes

$$yL_n'(y) = \sqrt{2n+1}\left(nP_n(z) + \sum_{k=0}^{n-1}(2k+1)P_k(z)\right).$$

(30) implies that $P_n(z) = \frac{L_n(y)}{\sqrt{2n+1}}$, thus

$$yL_n'(y) = nL_n(z) + \sqrt{2n+1}\left(\sum_{k=0}^{n-1}\sqrt{2k+1}L_k(z)\right).$$

$\square$

We now re-state and prove Corollary 3.1:

**Corollary C.8** (Corollary 3.1, restated)**.** *Let $L_n$ be the Legendre polynomials orthonormal over the interval $[0,1]$. Define $\sigma(t,s) = \exp(a(s) - a(t))$. The SSM $(a'(t)\boldsymbol{A}, a'(t)\boldsymbol{B})$ is an OSSM with*
$$\omega(t,s) = \mathbb{I}(\sigma(t,s))a'(s)\sigma(t,s) \qquad p_n(t,s) = L_n(\sigma(t,s)),$$
*where $\boldsymbol{A}$ and $\boldsymbol{B}$ are defined as in* (4).

*Proof.* We consider our basis functions, the Legendre polynomials, which are orthogonal with respect to unit measure. This allows us to invoke Corollary C.4 with $\omega = 1$. Further, here we have $t_0 = -\infty$ and $\eta = 0$. Now we have an SSM:
$$\left(-a'(t)(\boldsymbol{A}^0 + \boldsymbol{I}), a'(t)\boldsymbol{B}\right)$$
where $\boldsymbol{A}_{nk}^0 = \alpha_{nk}$ as in (26) and $\boldsymbol{B}_n = L_n(1)$.

From (29) observe that $\boldsymbol{B}_n = (2n+1)^{\frac{1}{2}}$. From (27), we have

$$\alpha_{nk} = \begin{cases} (2n+1)^{\frac{1}{2}}(2k+1)^{\frac{1}{2}} & k < n \\ n & k = n \\ 0 & \text{otherwise} \end{cases}.$$

We write that $\boldsymbol{A} = -(\boldsymbol{A}^0 + \boldsymbol{I})$. Indeed,

$$-(\boldsymbol{A}^0 + \boldsymbol{I})_{nk} = -\begin{cases} (2n+1)^{\frac{1}{2}}(2k+1)^{\frac{1}{2}} & \text{if } k < n \\ n+1 & \text{if } k = n \\ 0 & \text{if } k > n \end{cases}.$$

Thus the $\boldsymbol{A}$ and $\boldsymbol{B}$ match those in (4), which completes our claim. $\square$

We now re-state and prove Theorem 3:

**Corollary C.9** (Theorem 3, restated)**.** *Let $L_n$ be the Legendre polynomials orthonormal over the interval $[0,1]$. Then the SSM $(\frac{1}{t}\boldsymbol{A}, \frac{1}{t}\boldsymbol{B})$ is a OSSM for basis functions $L_n(\frac{s}{t})$ and measure $\frac{1}{t}\mathbb{I}[t_0,t]$ where $\boldsymbol{A}$ and $\boldsymbol{B}$ are defined as in* (4).

*Proof.* We consider our basis functions, the Legendre polynomials, which are orthogonal with respect to unit measure. This allows us to invoke Corollary C.5 with $\omega = 1$. Now we have

$$x'(t) = \frac{1}{t}\left[-(\boldsymbol{A}^0 + \boldsymbol{I})x(t) + \boldsymbol{B}u(t)\right]$$

where $\boldsymbol{A}_{nk}^0 = \alpha_{nk}$ as in (26) and $\boldsymbol{B}_n = L_n(1)$.

From (29) observe that $\boldsymbol{B}_n = (2n+1)^{\frac{1}{2}}$. From (27), we have

$$\alpha_{nk} = \begin{cases} (2n+1)^{\frac{1}{2}}(2k+1)^{\frac{1}{2}} & k < n \\ n & k = n \\ 0 & \text{otherwise} \end{cases}.$$

We write that $\boldsymbol{A} = -(\boldsymbol{A}^0 + \boldsymbol{I})$. Indeed,

$$-(\boldsymbol{A}^0+\boldsymbol{I})_{nk}=-\begin{cases}(2n+1)^{\frac{1}{2}}(2k+1)^{\frac{1}{2}} & \text{if } k<n \\ n+1 & \text{if } k=n\,, \\ 0 & \text{if } k>n\end{cases}$$

which completes our claim.

$\square$

We now restate and prove Corollary 3.3.

**Corollary C.10** (Corollary 3.3, restated). *Let $L_n$ be the Legendre polynomials orthonormal over the interval $[0,1]$. Then the SSM $(\boldsymbol{A},\boldsymbol{B})$ is a TOSSM for basis functions $L_n(e^{-t})$ with measure $\omega=\mathbb{I}[t_0,t]e^{-t}$ where $\boldsymbol{A},\boldsymbol{B}$ are defined as in* (4).

*Proof.* We consider our basis functions, the Legendre polynomials, which are orthogonal with respect to unit measure, warping function $\sigma=\exp(s-t)$, and with tilting $\chi=\omega$. We note that $\sigma=\exp(s-t)$ satisfies (24) with, $a'(t)=1$. This allows us to invoke Corollary C.5.

Then $x'(t) = (\boldsymbol{A} + \boldsymbol{I})x(t) + \boldsymbol{B}u(t)$ orthogonalizes against the basis functions $L_n(e^{s-t})$ with measure $\mathbb{I}[-\infty,t]e^{s-t}$ where $\boldsymbol{A} = \alpha_{nk}$ as in 26. Note that the SSM basis functions $K_n(t,s)=K_n(s-t)$, hence we get the claimed SSM form utilizing the same argument for $\boldsymbol{A},\boldsymbol{B}$ as in the proof of Corollary C.9 $\square$

This explains why removing the $\frac{1}{t}$ factor from HiPPO-LegS still works: it is orthogonalizing onto the Legendre polynomials with an exponential "warping".

## C.4 FINITE WINDOWS

### C.4.1 LEGT DERIVATION

**Corollary C.11.** *Let $L_n$ be the Legendre polynomials orthonormal over the interval $[0,1]$ and let $\sigma=1-\frac{t-s}{\theta}$ for a constant $\theta$. Then the SSM $\left(\frac{1}{\theta}\boldsymbol{A},\frac{1}{\theta}\boldsymbol{B}\right)$ is a OSSM for basis functions $L_n(\sigma)$ with measure $\frac{1}{\theta}\mathbb{I}[t_0,t](\sigma)$ where $\boldsymbol{A},\boldsymbol{B}$ are defined as in* (5).

*Proof.* Out plan is to apply Corollary C.3, for which we must show that the basis functions $L_n(t,s)$, time warping $\sigma(t,s)$, and tilting $\chi(t,s)=\psi^{-1}\phi^{-1}(t,s)$ satisfy (18), (15), and (19), respectively. We first set some parameters– note that because $\omega=1$ and set $\psi=\phi=1$.

The above implies that we have

$$\sigma_t(L_n\omega\psi)'(\sigma)=-\frac{1}{\theta}L_n'(\sigma).$$

The above along with (28), we see that the Legendre polynomials satisfy (18) with

$$\gamma_{nk}=\frac{1}{\theta}\cdot\begin{cases}-2\cdot(2n+1)^{\frac{1}{2}}(2k+1)^{\frac{1}{2}} & k<n \text{ and } n-k \text{ is odd} \\ 0 & \text{otherwise}\end{cases}. \tag{31}$$

We also note that $\sigma_s=\frac{1}{\theta}$.

It follows that

$$\frac{d}{dt}\sigma_s=0,$$

satisfying (15) trivially by setting $c(t)=0$. Similarly, since $\phi=1$ (19) is also satisfied trivially by setting $d(t)=0$. Finally we note that the $L_n$ forms a complete basis over $[0,1]$, hence as $N\to\infty$, we have

$$u(t-\theta)=\sum_{k=0}^{N-1}x_k(t)L_n(\sigma(t,t-\theta))=\sum_{k=0}^{N-1}x_k(t)L_n(0).$$

The above defines $\boldsymbol{A}'$ by setting $\boldsymbol{A}'_n=L_n(0)$ (as well as $\bar{c}=1$ and $\bar{d}=0$.) Now by Corollary C.3, we have an SSM

$$\left(\boldsymbol{A}^0 - \boldsymbol{D}(\boldsymbol{A'})^\top, \boldsymbol{B'}\right),$$

where $\boldsymbol{D}_n = \frac{1}{\theta} L_n(0)$, and by (27) $\boldsymbol{A}^0_{nk} = \gamma_{nk}$ (as in (31)) and $\boldsymbol{B'}_n = \frac{1}{\theta} L_n(1)$.

From (29), we have $\boldsymbol{D}_n = \frac{1}{\theta}(2n+1)^{\frac{1}{2}}(-1)^n$ and $\boldsymbol{B}_n = \frac{1}{\theta}(2n+1)^{\frac{1}{2}}$.

Thus, we have

$$\left(\boldsymbol{A}^0 - \boldsymbol{D}(\boldsymbol{A'})^\top\right)_{nk} = \frac{1}{\theta} \cdot \begin{cases} -(2n+1)^{\frac{1}{2}}(2k+1)^{\frac{1}{2}}\left(2 + (-1)^{n-k}\right) & k < n \text{ and } n-k \text{ is odd} \\ -(2n+1)^{\frac{1}{2}}(2k+1)^{\frac{1}{2}}(-1)^{n-k} & \text{otherwise} \end{cases}.$$

The proof is complete by noting that $\boldsymbol{A}^0 - \boldsymbol{D}(\boldsymbol{A'})^\top = \frac{1}{\theta}\boldsymbol{A}$ and $\boldsymbol{B'} = \frac{1}{\theta}\boldsymbol{B}$.

$\square$

We note that Corollary C.11 implies Proposition 2. More specifically, Proposition 2 follows by setting $\theta = 1$ in Corollary C.11 and noticing that the OSSM there is actually a TOSSM. (Technically we get basis function $L_n(1-t)$ for measure $\mathbb{I}(1-t)$ but this is OK since $\int_0^1 L_k(1-t)L_j(1-t)dt = \int_0^1 L_k(t)L_j(t)dt$.)

We first give a proof of Theorem 4. Then, we prove Theorem 9 as a function approximation result pertaining to S4-FouT.

### C.4.2   EXPLANATION OF S4-FOUT

*Proof of Theorem 4.* We seek to derive $\boldsymbol{A}$ and $\boldsymbol{B'}$ from (6) using Corollary C.3:

We use the time-warping function $\sigma(t,s) = 1 - (t-s)$, which implies that we have

$$\sigma_s(t,s) = 1, \tag{32}$$

$$\frac{\partial}{\partial t}\sigma_s(t,s) = 0 \tag{33}$$

Thus, we can take

$$c(t) = 0 \text{ in } \frac{\partial}{\partial t}\sigma_s(t,s) = c(t)\sigma_s(t,s). \tag{34}$$

We then have $\chi(t,s) = 1$ as we set

$$\psi(t,s) = \phi(t,s) = 1, \tag{35}$$

$$\frac{d}{dt}\phi(t,s) = 0. \tag{36}$$

So, we can take

$$d(t) = 0 \text{ in } \frac{d}{dt}\phi(t,s) = d(t)\phi(t,s). \tag{37}$$

We also have $\omega(\sigma) = 1$, and we order our bases in the form $p_n = (1, c_1(t), s_1(t), c_2(t), s_2(t), ...)$[3], where the basis functions have derivatives:

$$(1)'(\sigma) = 0;$$
$$(c_n)'(\sigma) = -2\pi n s_n(\sigma);$$
$$(s_n)'(\sigma) = 2\pi n c_n(\sigma).$$

Consequently, we can define $\gamma_{nk}$ as follows:

$$\gamma_{nk} = \begin{cases} 2\pi n & n-k = 1, \ k \text{ odd} \\ -2\pi k & k-n = 1, \ n \text{ odd}. \\ 0 & \text{otherwise} \end{cases} \tag{38}$$

Further, the discontinuity is at $t_0 = t - \theta$, $\theta = 1$ which implies that $\frac{dt_0}{dt} = 1$. We now seek to use the stored approximation to $u$ at time $t$ to compute $u(t-1)$.

---

[3]Note that this is 0-indexed.

First, denote the latent state $x(t)$ with coefficients $x = (x^1(t), x_1^c(t), x_1^s(t), x_2^c(t), x_2^s(t), ...)$ and define the functions $v(s)$ and $w(s)$ such that we have

$$v(s) = u(2t - s - 1) \quad \text{and} \quad w(s) = \frac{u(s) + v(s)}{2} \quad \text{for} \quad s \in [t-1, t].$$

Now, let $\hat{u}, \hat{v},$ and $\hat{w}$ denote the reconstruction of $u, v$ and $w$, where we have

$$\hat{u}(t,s) = \langle u(s), 1 \rangle + \sum_n \langle u(s), s_n(\sigma(t,s)) \rangle s_n(\sigma(t,s)) + \sum_n \langle u(s), c_n(\sigma(t,s)) \rangle c_n(\sigma(t,s)), \quad (39)$$

$$\hat{v}(t,s) = \langle v(s), 1 \rangle + \sum_n \langle v(s), s_n(\sigma(t,s)) \rangle s_n(\sigma(t,s)) + \sum_n \langle v(s), c_n(\sigma(t,s)) \rangle c_n(\sigma(t,s)), \quad (40)$$

$$\hat{w}(t,s) = \langle w(s), 1 \rangle + \sum_n \langle w(s), s_n(\sigma(t,s)) \rangle s_n(\sigma(t,s)) + \sum_n \langle w(s), c_n(\sigma(t,s)) \rangle c_n(\sigma(t,s)). \quad (41)$$

Then, we claim that

$$\hat{u}(t,t) = \frac{u(t) + v(t)}{2} = \frac{u(t) + u(t-1)}{2}. \quad (42)$$

Towards that end, we examine the sine and cosine coefficients of $u$ and $v$ as follows:

$$\langle v, c_n \rangle = \int v(s) c_n(\sigma(t,s)) \mathbb{I}[t-1,t] ds = \int u(2t-s-1) c_n(\sigma(t,s)) \mathbb{I}[t-1,t] ds$$

$$= \int u(s') c_n(1 - \sigma(t,s')) \mathbb{I}[t-1,t] ds' \quad (43)$$

$$= \int u(s') c_n(\sigma(t,s')) \mathbb{I}[t-1,t] ds' = \langle u, c_n \rangle.$$

$$\langle v, s_n \rangle = \int v(s) s_n(\sigma(t,s)) \mathbb{I}[t-1,t] ds = \int u(2t-s-1) s_n(\sigma(t,s)) \mathbb{I}[t-1,t] ds$$

$$= \int u(s') s_n(1 - \sigma(t,s')) \mathbb{I}[t-1,t] ds' \quad (44)$$

$$= -\int u(s') s_n(\sigma(t,s')) \mathbb{I}[t-1,t] ds' = -\langle u, s_n \rangle.$$

Here, for (43) and (44), we use the change of variables $s' \leftarrow 2t - s - 1$, which gives us

$$\sigma(t,s) = 1 - (t-s) = 1 - (1 + t - s - 1) = 1 - [1 - (t - (2t-s-1))] = 1 - (1 - (t-s')) = 1 - \sigma(t,s').$$

Then, we use the fact that $c_n(1 - \sigma(t,s')) = c_n(\sigma(t,s'))$ but $s_n(1 - \sigma(t,s')) = -s_n(\sigma(t,s'))$. That is, both $u$ and $v$ have the same cosine coefficients but negated sine coefficients of each other. But, we know that both $s_n(\sigma(t,t-1)) = s_n(1 - (t - (t-1))) = s_n(0) = 0$ and $s_n(\sigma(t,t)) = s_n(1 - (t-t)) = s_n(1) = 0$, and hence, the reconstruction of $\hat{u}$ at the endpoints $\sigma(t,t-1) = 0$ and $\sigma(t,t) = 1$ depends only on the cosine coefficients, whence we assert that the reconstruction $\hat{u}$ agrees with $\hat{v}$ at both endpoints. Therefore, we have $\hat{u}(t,t) = \hat{v}(t,t)$ implying that $\hat{w}(t,t) = \hat{u}(t,t)$.

Note that $w$ is continuous and periodic, for which the basis $\{1, c_n, s_n\}_n$ is complete, and hence, we know that as $N \to \infty$, $\hat{w} \to w$. Thus, at $s = t$, we have $\hat{u}(t,t) = \hat{w}(t,t) = w(t) = \frac{u(t)+v(t)}{2} = \frac{u(t)+u(t-1)}{2}$, which completes the proof of the claim in (42).

Recall from (39) that we can express the stored approximation of $u(t)$, given by $\hat{u}(t,s)$, as follows:

$$\hat{u}(t,s) = \langle u(s), 1 \rangle + \sum_n \langle u(s), s_n(\sigma(t,s)) \rangle s_n(\sigma(t,s)) + \sum_n \langle u(s), c_n(\sigma(t,s)) \rangle c_n(\sigma(t,s))$$

For the value at $t$, the approximation $\hat{u}(t,t)$ is then given by

$$\hat{u}(t,t) = x^1(t) + \sum_k x_k^c(t) c_k(1) + \sum_k x_k^s(t) s_k(1) = x^1(t) + \sum_k \sqrt{2} x_k^c(t).$$

Due to (42), we know $u(t-1) = 2\hat{u}(t,t) - u(t)$, which combined with the above yields:

$$u(t-1) = 2x^1(t) + 2\sqrt{2} \sum_k x_k^c(t) - u(t). \quad (45)$$

Finally, with regards to Corollary C.3, for Theorem 8, (34) satisfies (15) and (37) satisfies (19) with (38) satisfying (18) for $\boldsymbol{A}^0$. Moreover, from (45), we can take $\bar{c}=1, \bar{d}=-1$, and

$$\boldsymbol{A}'_k := \begin{cases} 2 & k=0 \\ 2\sqrt{2} & k \text{ odd} \\ 0 & \text{otherwise} \end{cases} \qquad \text{to satisfy (20).}$$

Invoking Corollary C.3 now yields the following OSSM:[4]

$$(\boldsymbol{A}^0 + (c(t)+d(t))\boldsymbol{I} - \bar{c}\boldsymbol{D}(\boldsymbol{A}')^\top, \ \boldsymbol{B}-\bar{d}\boldsymbol{D}),$$

where $\boldsymbol{A}^0_{nk} = \gamma_{nk}$ with $\boldsymbol{D}_n$ and $\boldsymbol{B}_n$ specified as follows:

$$\boldsymbol{D}_n = \begin{cases} 1 & n=0 \\ \sqrt{2} & n \text{ odd} \\ 0 & \text{otherwise} \end{cases} \tag{46}$$

$$\boldsymbol{B}_n = \begin{cases} 1 & n=0 \\ \sqrt{2}, & n \text{ odd} \\ 0 & \text{otherwise} \end{cases} \tag{47}$$

Here, the values are derived from the expressions of Corollary C.3:

$$\boldsymbol{D}_n = (p_n \omega \psi)(\sigma(t,t-1))(\sigma_s \phi)(t,t-1) \text{ and } \boldsymbol{B}_n = (p_n \omega \psi)(1)(\sigma_s \phi)(t,t).$$

Recall that we have $p_n \in \{1, c_n, s_n\}, \omega(t,s)=1$, and from (32) and (35), $\sigma_s(t,s)=1$ with $\psi(t,s)=\phi(t,s)=1$. Thus, (46) is due to $1(0)\cdot 1 = 1, s_n(0)\cdot 1 = 0$ but $c_n(0)\cdot 1 = \sqrt{2}$. Similarly, (47) is because $1(0)\cdot 1 = 1, s_n(1)\cdot 1 = 0$ but again $c_n(1)\cdot 1 = \sqrt{2}$.

Now, we have

$$[\boldsymbol{D}(\boldsymbol{A}')^\top]_{nk} = \begin{cases} 2 & n=k=0 \\ 2\sqrt{2} & n=0, \ k \text{ odd or } k=0, \ n \text{ odd} \\ 4 & n,k \text{ odd} \\ 0 & \text{otherwise} \end{cases} \qquad [\bar{d}\boldsymbol{D}]_n = \begin{cases} -1 & n=0 \\ -\sqrt{2} & n \text{ odd} \\ 0 & n \text{ otherwise} \end{cases}.$$

As $c(t)=d(t)=0$, we define $\boldsymbol{A} \leftarrow \boldsymbol{A}^0 - \bar{c}\boldsymbol{D}(\boldsymbol{A}')^\top$ and $\boldsymbol{B} \leftarrow \boldsymbol{B}-\bar{d}\boldsymbol{D}$, given by

$$\boldsymbol{A}_{nk} = \begin{cases} -2 & n=k=0 \\ -2\sqrt{2} & n=0, \ k \text{ odd or } k=0, \ n \text{ odd} \\ -4 & n,k \text{ odd} \\ 2\pi n & n-k=1, k \text{ odd} \\ -2\pi k & k-n=1, n \text{ odd} \\ 0 & \text{otherwise} \end{cases} \qquad \boldsymbol{B}_n = \begin{cases} 2 & n=0 \\ 2\sqrt{2} & n \text{ odd} \\ 0 & \text{otherwise} \end{cases}$$

$\square$

### C.4.3    Function Approximation Error

**Theorem 9.** *Let $K(t)$ be a differentiable kernel on $[0,1]$, and let $\hat{K}(t)$ be its representation by the FouT system (Theorem 4) with state size $N$. If $K$ is $L-$Lipschitz, then for $\epsilon > 0, N \geq \left(\frac{L}{\pi\epsilon}\right)^2 + 2$, we have $\|K(t)-\hat{K}(t)\| \leq \epsilon$. If $K$ has $k-$derivatives bounded by $L$, then we can take $N \geq \left(\frac{L}{\pi^k \epsilon}\right)^{\frac{2}{2k-1}} + 2$.*

*Proof of Theorem 9.* First, the state size being $N$ dictates that there are $\lfloor N/2 \rfloor$ $s_n$ and $c_n$ basis functions each. We fix time $t$ and denote $x_n^c$ and $x_n^s$ to be the respective coefficients for $s_n$ and $c_n$ basis corresponding to S4-Fou. Since $\{s_n, c_n\}_{n\geq 0}$ forms an orthonormal basis, by Parseval's identity, we have

$$\|K-\hat{K}\|_2^2 = \sum_{n=\lfloor N/2 \rfloor}^{\infty} x_n^{c\,2}(t) + x_n^{s\,2}(t). \tag{48}$$

---

[4]Recall that, like the coefficients, the matrices are 0-indexed.

Thus, in order to bound the error, it suffices to bound the high-order coefficients by integration by parts as follows:

$$x_n^c(t) = \langle K, c_n \rangle = \int_0^1 K(t) c_n(t) dt$$

$$= \left[ K(t) \frac{1}{2\pi n} s_n(t) \right]_0^1 - \frac{1}{2\pi n} \int_0^1 K'(t) s_n(t) dt$$

$$= -\frac{1}{2\pi n} \int_0^1 K'(t) s_n(t) dt.$$

The quantity in the bracket vanishes as $s_n$ is periodic. Therefore

$$|x_n^c| \leq \left| \frac{1}{2\pi n} \int_0^1 K'(t) s_n(t) dt \right| \leq \frac{1}{2\pi n} \int_0^1 |K'(t)| |s_n(t)| dt \leq \frac{L}{2\pi n},$$

where we use the fact that $K$ is $L-$Lipshitz. For $x_n^s$, a similar argument holds and we get:

$$|x_n^s| \leq \left| \frac{1}{2\pi} \int_0^1 K'(t) c_n(t) dt \right| \leq \frac{1}{2\pi} \int_0^1 |K'(t)| |c_n(t)| dt \leq \frac{L}{2\pi n}.$$

Due to (48), this then implies that

$$\|K - \hat{K}\|_2^2 = \sum_{n=\lfloor N/2 \rfloor}^\infty x_n^{c\,2}(t) + x_n^{s\,2}(t) = \sum_{n=\lfloor N/2 \rfloor}^\infty |x_n^c|^2(t) + |x_n^s|^2(t)$$

$$\leq \sum_{n=\lfloor N/2 \rfloor}^\infty \frac{2L^2}{(2\pi n)^2} = \frac{2L^2}{(2\pi)^2} \sum_{n=\lfloor N/2 \rfloor}^\infty \frac{1}{n^2} = \frac{2L^2}{(2\pi)^2} \frac{1}{\lfloor N/2 \rfloor}$$

$$\leq \frac{L^2}{\pi^2(N-2)}. \tag{49}$$

We use (49) to get the following estimate on $\|K - \hat{K}\|$:

$$\|K - \hat{K}\|_2 \leq \frac{L}{\pi \sqrt{(N-2)}}.$$

Thus, it suffices for $N$ to satisfy the following inequality:

$$\frac{L}{\pi \sqrt{(N-2)}} \leq \epsilon \implies \sqrt{N-2} \geq \frac{L}{\pi \epsilon} \implies N \geq \left( \frac{L}{\pi \epsilon} \right)^2 + 2.$$

We now use the same argument as above to the fact that $K$ has order-$k$ bounded derivative. By iteration, we get:

$$|x_n^s| = |x_n^c| \leq \left| \frac{1}{(2\pi n)^k} \int_0^1 K^{(k)}(t) s_n(t) dt \right| \leq \frac{1}{(2\pi n)^k} \int_0^1 |K^{(k)}| |s_n(t)| dt \leq \frac{L}{(2\pi n)^k}.$$

Again, due to (48), this then gives us the following estimate on the square error:

$$\|K - \hat{K}\|_2^2 = \sum_{n=\lfloor N/2 \rfloor}^\infty x_n^{c\,2}(t) + x_n^{s\,2}(t) = \sum_{n=\lfloor N/2 \rfloor}^\infty |x_n^c|^2(t) + |x_n^s|^2(t)$$

$$\leq \sum_{n=\lfloor N/2 \rfloor}^\infty \frac{2L^2}{(2\pi n)^{2k}} = \frac{2L^2}{(2\pi)^{2k}} \sum_{n=\lfloor N/2 \rfloor}^\infty \frac{1}{n^{2k}} = \frac{2L^2}{(2\pi)^{2k}} \frac{1}{(\lfloor N/2 \rfloor)^{2k-1}}$$

$$\leq \frac{L^2}{\pi^{2k}(N-2)^{2k-1}}. \tag{50}$$

If $K$ has order $k-$bounded derivatives, then we use (50) to get the following estimate on $\|K - \hat{K}\|$:

$$\|K - \hat{K}\|_2 \leq \frac{L}{\pi^k(N-2)^{-k+1/2}}.$$

Again, it suffices for $N$ to satisfy the following inequality:

$$\frac{L}{\pi^k(N-2)^{-k+1/2}} \leq \epsilon \implies (N-2)^{k-1/2} \geq \frac{L}{\pi^k \epsilon} \implies N \geq \left( \frac{L}{\pi^k \epsilon} \right)^{\frac{2}{2k-1}} + 2.$$

$\square$

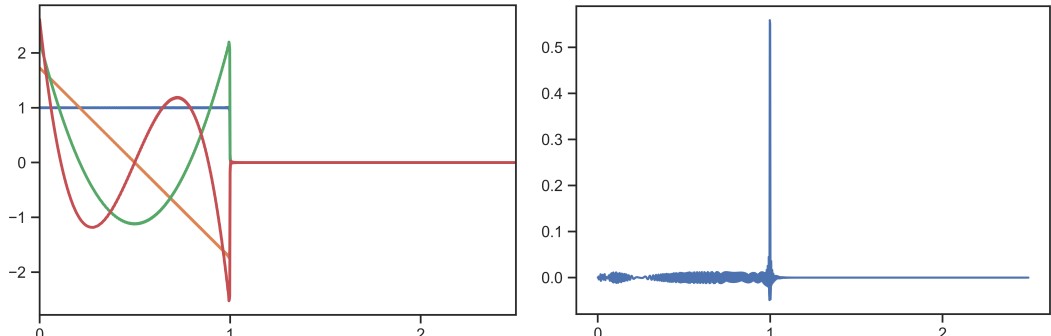

Figure 7: (**HiPPO-LegT.**) (*Left*) First 4 basis functions $K_n(t)$ for state size $N=1024$ (Proposition 2). (*Right*) Choosing a particular $C$ produces a spike kernel or "delay network" (Theorem 10).

### C.4.4    Approximating Delay Networks

The original motivation for the LDN/LMU (Voelker, 2019; Voelker et al., 2019) worked backward from the transfer function of the desired delay function impulse response $K(t)=\delta(t-1)$, and noticed that the SSM for Padé approximations to this were linked to Legendre polynomials. This was not fully proven, and we state it here and provide a full proof.

**Theorem 10.** *For $A,B,C,D$ in the LegT system described in Theorem 5, the transfer function $\mathcal{L}\{K(t)\}(s)$ is the $[N-1/N]$ Padé approximant to $e^{-s}=\mathcal{L}\{\delta(t-1)\}(s)$.*

We remark that although LegT (LMU) is designed to be an "optimal" approximation to the delay function via Padé approximants, it actually produces a weaker spike function than FouT (Fig. 7 vs. Fig. 1) and empirically performs slightly worse on synthetic tasks testing this ability (Section 4.3). This may be because Padé approximation in the Laplace domain does not necessarily translate to localization in the time domain.

Finally, we prove Theorem 10. Note that this is a stronger version of the LegT portion of Theorem 5, while the FouT portion is a corollary of the proof of Theorem 4.

We start by working out some calculations concretely to provide an example. The SSM corresponding to HiPPO-LegT is

$$A=P^{\frac{1}{2}}\begin{bmatrix}-1 & 1 & -1 & 1\\ -1 & -1 & 1 & -1\\ -1 & -1 & -1 & 1\\ -1 & -1 & -1 & -1\end{bmatrix}P^{\frac{1}{2}}$$

$$B=P^{\frac{1}{2}}\mathbf{1}$$

$$C=Z^{\top}P^{\frac{1}{2}}$$

$$P=\mathrm{diag}\{1+2n\}$$
$$Z^{\top}=\begin{bmatrix}1 & -1 & 1 & -1\end{bmatrix}$$

The transfer function is

$$C(sI-A)^{-1}B=Z(sP^{-1}-A)^{-1}\mathbf{1}$$

(In the RHS and for the rest of this part, we will redefine $A$ to be the $\pm 1$ matrix found above for convenience.)

**Case N=1.** We have $A=-1, B=C=1$, and the transfer function is $C(sI-A)^{-1}B=\frac{1}{1+s}$.

**Case N=2.** The transfer function is

$$C(sI-A)^{-1}B=\begin{bmatrix}1 & -1\end{bmatrix}\left(sP^{-1}-\begin{bmatrix}-1 & 1\\ -1 & -1\end{bmatrix}\right)^{-1}\begin{bmatrix}1\\ 1\end{bmatrix}$$

$$=\begin{bmatrix}1 & -1\end{bmatrix}\begin{bmatrix}s+1 & -1\\ 1 & \frac{s}{3}+1\end{bmatrix}^{-1}\begin{bmatrix}1\\ 1\end{bmatrix}$$

$$=\frac{1}{\frac{s^2}{3}+\frac{4s}{3}+2}\begin{bmatrix}1 & -1\end{bmatrix}\begin{bmatrix}1+\frac{s}{3} & 1\\ -1 & 1+s\end{bmatrix}\begin{bmatrix}1\\ 1\end{bmatrix}$$

$$=\frac{2-\frac{2s}{3}}{\frac{s^2}{3}+\frac{4s}{3}+2}$$

$$=\frac{1-\frac{s}{3}}{1+\frac{2s}{3}+\frac{s^2}{6}}$$

It can be verified that this is indeed $[1/2]_{\exp}(-s)$.

**A General Recursion.** We will now sketch out a method to relate these transfer functions recursively.

We will redefine $Z$ to be the vector that ENDS in $+1$.

The main idea is to write

$$A_n=\begin{bmatrix}A_{n-1} & Z_{n-1}\\ -\mathbf{1}_{n-1}^\top & -1\end{bmatrix}$$

$$\left(sP_n^{-1}-A_n\right)^{-1}=\begin{bmatrix}sP_{n-1}^{-1}-A_{n-1} & -Z_{n-1}\\ \mathbf{1}_{n-1}^\top & 1+\frac{s}{2n+1}\end{bmatrix}^{-1}.$$

Now we can use the block matrix inversion formula.[5] Ideally, this will produce a recurrence where the desired transfer function $Z_n(sP_n^{-1}-A_n)^{-1}\mathbf{1}_n$ will depend on $Z_{n-1}(sP_n^{-1}-A_{n-1})^{-1}\mathbf{1}_{n-1}$. However, looking at the block matrix inversion formula, it becomes clear that there are also dependencies on terms like $\mathbf{1}_{n-1}^\top(sP_{n-1}^{-1}-A_{n-1})^{-1}\mathbf{1}_{n-1}$ and $Z_{n-1}(sP_{n-1}^{-1}-A_{n-1})^{-1}Z_{n-1}^\top$.

The solution is to track all of these terms simultaneously.

We will compute the 4 transfer functions

$$H_n(s):=\begin{bmatrix}H_n^{1z}(s) & H_n^{11}(s)\\ H_n^{zz}(s) & H_n^{z1}(s)\end{bmatrix}$$

$$:=\begin{bmatrix}\mathbf{1}_n^\top(sP_n^{-1}-A_n)^{-1}Z_n & \mathbf{1}_n^\top(sP_n^{-1}-A_n)^{-1}\mathbf{1}_n\\ Z_n^\top(sP_n^{-1}-A_n)^{-1}Z_n & Z_n^\top(sP_n^{-1}-A_n)^{-1}\mathbf{1}_n\end{bmatrix}$$

$$=\begin{bmatrix}\mathbf{1}_n^\top\\ Z_n^\top\end{bmatrix}(sP_n^{-1}-A_n)^{-1}\begin{bmatrix}Z_n & \mathbf{1}_n\end{bmatrix}$$

**Lemma C.12.** *Instead of using the explicit block matrix inversion formula, it will be easier to work with the following factorization used to derive it (block LDU decomposition[6] ).*

$$\begin{bmatrix}A & B\\ C & D\end{bmatrix}=\begin{bmatrix}I & 0\\ CA^{-1} & I\end{bmatrix}\begin{bmatrix}A & 0\\ 0 & D-CA^{-1}B\end{bmatrix}\begin{bmatrix}I & A^{-1}B\\ 0 & I\end{bmatrix}$$

$$\begin{bmatrix}A & B\\ C & D\end{bmatrix}^{-1}=\begin{bmatrix}I & -A^{-1}B\\ 0 & I\end{bmatrix}\begin{bmatrix}A^{-1} & 0\\ 0 & (D-CA^{-1}B)^{-1}\end{bmatrix}\begin{bmatrix}I & 0\\ -CA^{-1} & I\end{bmatrix}$$

---

[5] https://en.wikipedia.org/wiki/Block_matrix#/Block_matrix_inversion
[6] https://en.wikipedia.org/wiki/Schur_complement#/Background

Using Lemma C.12, we can factor the inverse as

$$(s\boldsymbol{P}_n^{-1}-\boldsymbol{A}_n)^{-1} = \begin{bmatrix} \boldsymbol{I}_{n-1} & (s\boldsymbol{P}_{n-1}^{-1}-\boldsymbol{A}_{n-1})^{-1}\boldsymbol{Z}_{n-1} \\ & 1 \end{bmatrix}$$

$$\begin{bmatrix} (s\boldsymbol{P}_{n-1}^{-1}-\boldsymbol{A}_{n-1})^{-1} & \\ & \left(1+\frac{s}{2n+1}+(-1)^{n-1}\boldsymbol{H}_{n-1}^{1z}(s)\right)^{-1} \end{bmatrix}$$

$$\begin{bmatrix} \boldsymbol{I}_{n-1} & \\ -\boldsymbol{1}_{n-1}^{\top}(s\boldsymbol{P}_{n-1}^{-1}-\boldsymbol{A}_{n-1})^{-1} & 1 \end{bmatrix}$$

Now we compute

$$\begin{bmatrix} \boldsymbol{1}_n^{\top} \\ \boldsymbol{Z}_n^{\top} \end{bmatrix}\begin{bmatrix} \boldsymbol{I}_{n-1} & (s\boldsymbol{P}_{n-1}^{-1}-\boldsymbol{A}_{n-1})^{-1}\boldsymbol{Z}_{n-1} \\ & 1 \end{bmatrix}$$

$$= \begin{bmatrix} \boldsymbol{1}_{n-1}^{\top} & 1 \\ -\boldsymbol{Z}_{n-1}^{\top} & 1 \end{bmatrix}\begin{bmatrix} \boldsymbol{I}_{n-1} & (s\boldsymbol{P}_{n-1}^{-1}-\boldsymbol{A}_{n-1})^{-1}\boldsymbol{Z}_{n-1} \\ & 1 \end{bmatrix}$$

$$= \begin{bmatrix} \boldsymbol{1}_{n-1}^{\top} & 1+\boldsymbol{H}_{n-1}^{1z}(s) \\ -\boldsymbol{Z}_{n-1}^{\top} & 1-\boldsymbol{H}_{n-1}^{zz}(s) \end{bmatrix}$$

and

$$\begin{bmatrix} \boldsymbol{I}_{n-1} & \\ -\boldsymbol{1}_{n-1}^{\top}(s\boldsymbol{P}_{n-1}^{-1}-\boldsymbol{A}_{n-1})^{-1} & 1 \end{bmatrix}\begin{bmatrix} \boldsymbol{Z}_n & \boldsymbol{1}_n \end{bmatrix}$$

$$= \begin{bmatrix} \boldsymbol{I}_{n-1} & \\ -\boldsymbol{1}_{n-1}^{\top}(s\boldsymbol{P}_{n-1}^{-1}-\boldsymbol{A}_{n-1})^{-1} & 1 \end{bmatrix}\begin{bmatrix} -\boldsymbol{Z}_{n-1} & \boldsymbol{1}_{n-1} \\ 1 & 1 \end{bmatrix}$$

$$= \begin{bmatrix} -\boldsymbol{Z}_{n-1} & \boldsymbol{1}_{n-1} \\ 1+\boldsymbol{H}_{n-1}^{1z}(s) & 1-\boldsymbol{H}_{n-1}^{11}(s) \end{bmatrix}$$

Now we can derive the full recurrence for all these functions.

$$\boldsymbol{H}_n(s) = \begin{bmatrix} \boldsymbol{H}_n^{1z}(s) & \boldsymbol{H}_n^{11}(s) \\ \boldsymbol{H}_n^{zz}(s) & \boldsymbol{H}_n^{z1}(s) \end{bmatrix}$$

$$= \begin{bmatrix} \boldsymbol{1}_n^\top \\ \boldsymbol{Z}_n^\top \end{bmatrix} (s\boldsymbol{P}_n^{-1} - \boldsymbol{A}_n)^{-1} [\boldsymbol{Z}_n \quad \boldsymbol{1}_n]$$

$$= \begin{bmatrix} \boldsymbol{1}_n^\top \\ \boldsymbol{Z}_n^\top \end{bmatrix} \begin{bmatrix} \boldsymbol{I}_{n-1} & (s\boldsymbol{P}_{n-1}^{-1} - \boldsymbol{A}_{n-1})^{-1}\boldsymbol{Z}_{n-1} \\ & 1 \end{bmatrix}$$

$$\begin{bmatrix} (s\boldsymbol{P}_{n-1}^{-1} - \boldsymbol{A}_{n-1})^{-1} & \\ & \left(1 + \frac{s}{2n+1} + (-1)^{n-1}\boldsymbol{H}_{n-1}^{1z}(s)\right)^{-1} \end{bmatrix}$$

$$\begin{bmatrix} \boldsymbol{I}_{n-1} & \\ -\boldsymbol{1}_{n-1}^\top (s\boldsymbol{P}_{n-1}^{-1} - \boldsymbol{A}_{n-1})^{-1} & 1 \end{bmatrix} [\boldsymbol{Z}_n \quad \boldsymbol{1}_n]$$

$$= \begin{bmatrix} \boldsymbol{1}_{n-1}^\top & 1 + \boldsymbol{H}_{n-1}^{1z}(s) \\ -\boldsymbol{Z}_{n-1}^\top & 1 - \boldsymbol{H}_{n-1}^{zz}(s) \end{bmatrix}$$

$$\cdot \begin{bmatrix} (s\boldsymbol{P}_{n-1}^{-1} - \boldsymbol{A}_{n-1})^{-1} & \\ & \left(1 + \frac{s}{2n+1} + \boldsymbol{H}_{n-1}^{1z}(s)\right)^{-1} \end{bmatrix}$$

$$\cdot \begin{bmatrix} -\boldsymbol{Z}_{n-1} & \boldsymbol{1}_{n-1} \\ 1 + \boldsymbol{H}_{n-1}^{1z}(s) & 1 - \boldsymbol{H}_{n-1}^{11}(s) \end{bmatrix}$$

$$= \begin{bmatrix} \boldsymbol{1}_{n-1}^\top \\ -\boldsymbol{Z}_{n-1}^\top \end{bmatrix} (s\boldsymbol{P}_{n-1}^{-1} - \boldsymbol{A}_{n-1})^{-1} [-\boldsymbol{Z}_{n-1} \quad \boldsymbol{1}_{n-1}]$$

$$+ \begin{bmatrix} 1 + \boldsymbol{H}_{n-1}^{1z}(s) \\ 1 - \boldsymbol{H}_{n-1}^{zz}(s) \end{bmatrix} \left(1 + \frac{s}{2n+1} + \boldsymbol{H}_{n-1}^{1z}(s)\right)^{-1} [1 + \boldsymbol{H}_{n-1}^{1z}(s) \quad 1 - \boldsymbol{H}_{n-1}^{11}(s)]$$

$$= \begin{bmatrix} -\boldsymbol{H}_{n-1}^{1z}(s) & \boldsymbol{H}_{n-1}^{11}(s) \\ \boldsymbol{H}_{n-1}^{zz}(s) & -\boldsymbol{H}_{n-1}^{z1}(s) \end{bmatrix}$$

$$+ \begin{bmatrix} 1 + \boldsymbol{H}_{n-1}^{1z}(s) \\ 1 - \boldsymbol{H}_{n-1}^{zz}(s) \end{bmatrix} \left(1 + \frac{s}{2n+1} + \boldsymbol{H}_{n-1}^{1z}(s)\right)^{-1} [1 + \boldsymbol{H}_{n-1}^{1z}(s) \quad 1 - \boldsymbol{H}_{n-1}^{11}(s)]$$

Now we'll define a few transformations which will simplfy the calculations. Define

$$G_n^{1z}(s) = \frac{1}{2}(1 + \boldsymbol{H}_n^{1z}(s))$$
$$G_n^{11}(s) = 1 - \boldsymbol{H}_n^{1z}(s)$$
$$G_n^{zz}(s) = 1 - \boldsymbol{H}_n^{1z}(s)$$
$$G_n^{z1}(s) = (-1)^n \boldsymbol{H}_n^{z1}(s)$$

These satisfy the following recurrences:

$$G_n^{1z}(s) = 1 - G_{n-1}^{1z}(s) + \frac{G_{n-1}^{1z}(s) G_{n-1}^{1z}(s)}{G_{n-1}^{1z}(s) + \frac{s}{2(2n+1)}}$$

$$G_n^{11}(s) = G_{n-1}^{11}(s) - \frac{G_{n-1}^{11}(s) G_{n-1}^{1z}(s)}{G_{n-1}^{1z}(s) + \frac{s}{2(2n+1)}}$$

$$G_n^{zz}(s) = G_{n-1}^{zz}(s) - \frac{G_{n-1}^{zz}(s) G_{n-1}^{1z}(s)}{G_{n-1}^{1z}(s) + \frac{s}{2(2n+1)}}$$

$$G_n^{z1}(s) = G_{n-1}^{z1}(s) - (-1)^{n-1} \frac{G_{n-1}^{11}(s) G_{n-1}^{zz}(s)}{G_{n-1}^{1z}(s) + \frac{s}{2(2n+1)}}$$

We can analyze each term separately.

**Case $G_n^{1z}(s)$.**

This will be the most important term, as it determines the denominator of the expressions. Simplifying the recurrence slightly gives

$$G_n^{1z}(s) = 1 - G_{n-1}^{1z}(s) + \frac{G_{n-1}^{1z}(s)G_{n-1}^{1z}(s)}{G_{n-1}^{1z}(s) + \frac{s}{2(2n+1)}}$$

$$= \frac{(G_{n-1}^{1z}(s) + \frac{s}{2(2n+1)}) - \frac{s}{2(2n+1)} \cdot G_{n-1}^{1z}(s)}{G_{n-1}^{1z}(s) + \frac{s}{2(2n+1)}}.$$

Now let $G_n^{1z}(s) = \frac{P_n^{1z}(s)}{Q_n^{1z}(s)}$ where $P, Q$ are polynomials. Clearing the denominator $Q$ yields

$$\frac{P_n^{1z}(s)}{Q_n^{1z}(s)} = \frac{(P_{n-1}^{1z}(s) + \frac{s}{2(2n+1)}Q_{n-1}^{1z}(s)) - \frac{s}{2(2n+1)} \cdot P_{n-1}^{1z}(s)}{P_{n-1}^{1z}(s) + \frac{s}{2(2n+1)} \cdot Q_{n-1}^{1z}(s)}.$$

This results in the recurrence

$$Q_n^{1z}(s) = P_{n-1}^{1z}(s) + \frac{s}{2(2n+1)} \cdot Q_{n-1}^{1z}(s)$$

$$P_n^{1z}(s) = Q_{n-1}^{1z}(s) - \frac{s}{2(2n+1)} \cdot P_{n-1}^{1z}(s).$$

But this is exactly the *fundamental recurrence formula* for continuants of the continued fraction

$$e^s = 1 + \cfrac{s}{1 - \cfrac{\frac{1}{2}s}{1 + \cfrac{\frac{1}{6}s}{1 - \cfrac{\frac{1}{6}s}{1 + \cfrac{\frac{1}{10}s}{1 - \cfrac{\frac{1}{10}s}{1 + \ddots}}}}}}$$

Therefore $Q_{n-1}^{1z}(s)$ are the denominators of the Pade approximants.

Note that by definition of $P, Q$,

$$G_{n-1}^{1z}(s) + \frac{s}{2(2n+1)} = \frac{P_{n-1}^{1z}(s) + \frac{s}{2(2n+1)} \cdot Q_{n-1}^{1z}(s)}{Q_{n-1}^{1z}(s)} = \frac{Q_n^{1z}(s)}{Q_{n-1}^{1z}(s)}$$

Going forward we will also suppress the superscript of $Q$, $Q_{n-1}(s) := Q_{n-1}^{1z}(s)$, as it will be evident that all terms have the same denominator $Q_n(s)$

**Case $G_n^{11}(s)$.**

First note that $G_n^{11}(s) = G_n^{zz}(s)$ is straightforward from the fact that their recurrences are identical. The recurrence is

$$G_n^{11}(s) = G_{n-1}^{11}(s) - \frac{G_{n-1}^{11}(s)G_{n-1}^{1z}(s)}{G_{n-1}^{1z}(s) + \frac{s}{2(2n+1)}}$$

$$= \frac{\frac{s}{2(2n+1)} \cdot G_{n-1}^{11}(s)}{G_{n-1}^{1z}(s) + \frac{s}{2(2n+1)}}$$

$$= \frac{\frac{s}{2(2n+1)} \cdot G_{n-1}^{11}(s)Q_{n-1}(s)}{P_{n-1}^{1z}(s) + \frac{s}{2(2n+1)} \cdot Q_{n-1}(s)}$$

$$= \frac{\frac{s}{2(2n+1)} \cdot G_{n-1}^{11}(s)Q_{n-1}(s)}{Q_n(s)}$$

Therefore

$$G_n^{11}(s)Q_n(s) = \frac{s}{2(2n+1)} \cdot G_{n-1}^{11}(s)Q_{n-1}(s) = \prod_{i=1}^{n} \frac{s}{2(2i+1)}$$

$$G_n^{11}(s) = \frac{\prod_{i=1}^{n} \frac{s}{2(2i+1)}}{Q_n(s)}$$

**Case $G_n^{z1}(s)$.**

Define

$$G_n^{z1}(s) = \frac{P_n^{z1}(s)}{Q_n(s)}$$

This term satisfies the formula

$$G_n^{z1}(s) = G_{n-1}^{z1}(s) - (-1)^{n-1} \frac{G_{n-1}^{11}(s)G_{n-1}^{zz}(s)}{G_{n-1}^{1z}(s) + \frac{s}{2(2n+1)}}$$

$$= \frac{P_{n-1}^{z1}(s)}{Q_{n-1}(s)} - (-1)^{n-1} \frac{\left(\prod_{i=0}^{n-1} \frac{s}{2(2i+1)}\right)^2 / Q_{n-1}(s)^2}{\frac{Q_n(s)}{Q_{n-1}(s)}}$$

$$= \frac{P_{n-1}^{z1}(s)Q_n(s)}{Q_{n-1}(s)Q_n(s)} - (-1)^{n-1} \frac{\left(\prod_{i=0}^{n-1} \frac{s}{2(2i+1)}\right)^2}{Q_n(s)Q_{n-1}(s)}$$

By definition of $P^{z1}$,

$$P_{n-1}^{z1}(s)Q_n(s) - (-1)^{n-1}\left(\prod_{i=0}^{n-1} \frac{s}{2(2i+1)}\right)^2 = P_n^{z1}(s)Q_{n-1}(s)$$

But note that this is exactly satisfied by the Padé approximants, by the determinantal formula of continued fractions. This shows that $G_{n-1}^{1z}(s)$ are the Padé approximants of $e^{-s}$, as desired.

## C.5  Normalization and Timescales

**Proposition 11** (Closure properties of TOSSMs)**.** *Consider a TOSSM $(\boldsymbol{A},\boldsymbol{B})$ for basis functions $p_n(t)$ and measure $\omega(t)$. Then, the following are also TOSSMs with the corresponding basis functions and measure:*

1. *Constant scaling changes the timescale: $(c\boldsymbol{A},c\boldsymbol{B})$ is a TOSSM with basis $p(ct)$ and measure $\omega(ct)c$.*

2. *Identity shift tilts by exponential: $(\boldsymbol{A}+c\boldsymbol{I},\boldsymbol{B})$ is a TOSSM with basis $p(t)e^{-ct}$ and measure $\omega(t)e^{2ct}$.*

3. *Unitary change of basis preserves measure: $(\boldsymbol{V}\boldsymbol{A}\boldsymbol{V}^*,\boldsymbol{V}\boldsymbol{B})$ is a TOSSM with basis $\boldsymbol{V}p(t)$ and measure $\omega(t)$.*

*Proof.* We define $p(t)$ to be the vector of basis functions for the OSSM $(\boldsymbol{A},\boldsymbol{B})$,

$$p(t) = \begin{bmatrix} p_0(t) \\ \vdots \\ p_{N-1}(t) \end{bmatrix}.$$

Recall that the SSM kernels are $K_n(t) = p_n(t)\omega(t)$ so that $p(t)\omega(t) = e^{t\boldsymbol{A}}\boldsymbol{B}$.

1. The SSM kernels are

$$e^{t(c\boldsymbol{A})}(c\boldsymbol{B}) = ce^{(ct)\boldsymbol{A})}\boldsymbol{B} = cp(ct)\omega(ct).$$

It remains to show that the $p_n(ct)$ are orthonormal with respect to measure $c\omega(ct)$:

$$\int p_j(ct)p_k(ct)\omega(ct)c = \delta_{jk}$$

which follows immediately from the change of variables formula.

2. Using the commutativity of $\boldsymbol{A}$ and $\boldsymbol{I}$, the SSM kernels are

$$e^{t(\boldsymbol{A}+c\boldsymbol{I})}\boldsymbol{B} = e^{t\boldsymbol{A}}e^{ct\boldsymbol{I}}\boldsymbol{B} = e^{ct}p(t)\omega(t).$$

It remains to show that $p_n(t)e^{-ct}$ are orthonormal with respect to measure $\omega(t)e^{2ct}$:

$$\int p_j(t)e^{-ct}p_k(t)e^{-ct}\omega(t)e^{2ct} = \int p_j(t)p_k(t)\omega(t) = \delta_{jk}.$$

3. The SSM basis is

$$e^{t\boldsymbol{V}\boldsymbol{A}\boldsymbol{V}^*}\boldsymbol{V}\boldsymbol{B} = \boldsymbol{V}e^{t\boldsymbol{A}}\boldsymbol{B} = \boldsymbol{V}p(t)\omega(t).$$

It remains to show that the basis functions $\boldsymbol{V}p(t)$ are orthonormal with respect to $\omega(t)$. Note that orthonormality of a set of basis functions can be expressed as $\int p(t)\omega(t)p(t)^\top = \boldsymbol{I}$, so that

$$\int (\boldsymbol{V}p(t))\omega(t)(\boldsymbol{V}p(t))^* = \boldsymbol{V}\left[\int p(t)\omega(t)p(t)^\top\right]\boldsymbol{V}^*$$
$$= \boldsymbol{I}.$$

$\square$

**Normalization.** A standard aspect of training deep learning models, in general, concerns the scale or variance of activations. This has been the subject of much research on training deep learning models, touching on deep learning theory for the dynamics of training such as the exploding/vanishing gradient problem (Hochreiter, 1991), and a large number of normalization methods to ensure properly normalized methods, from the simple Xavier/He initializations (Glorot and Bengio, 2010; He et al., 2015) to BatchNorm and LayerNorm (Ioffe and Szegedy, 2015; Ba et al., 2016) to many modern variants and analyses of these (Davis et al., 2021).

The following proposition follows because for a TOSSM, $x(t)$ can be interpreted as projecting onto orthonormal functions in a Hilbert space (Proposition 1).

**Proposition 12** (Normalization of TOSSM). *Consider an (infinite-dimensional) TOSSM. For any input $u(t)$, $\|x(t)\|_2^2 = \|u\|_\omega^2 = \int_{-\infty}^t u(s)^2 \omega(t-s)dt$.*

**Corollary C.13.** *For a TOSSM with a probability measure (i.e. $\int \omega(t) = 1$) and any constant input $u(t) = c$, the state has norm $\|x(t)\|^2 = c^2$ and the output $y(t)$ has mean 0, variance $c^2$ if the entries of $\boldsymbol{C}$ are mean $0$ and variance $1$.*

Note that the probability measure requirement can be satisfied by simply rescaling $\boldsymbol{B}$. Corollary C.13 says the TOSSM preserves the variance of inputs, the critical condition for a properly normalized deep learning layer. Note that the initialization of $\boldsymbol{C}$ is different than a standard Linear layer in deep neural networks, which usually rescale by factor depending on its dimensionality such as $N^{-\frac{1}{2}}$ (Glorot and Bengio, 2010).

