# OpenReview forum: "How to Train your HIPPO: State Space Models with Generalized Orthogonal Basis Projections"
_ICLR.cc/2023/Conference — ICLR 2023 poster_

### Official Review · Reviewer_ZKrd · 2022-10-23

**Confidence:** 3
**Correctness:** 4
**Technical Novelty And Significance:** 4
**Empirical Novelty And Significance:** Not applicable
**Recommendation:** 8

**Clarity, Quality, Novelty And Reproducibility:**

The paper is mostly clear, and the quality of the research is high. The authors provide some novelty with the new analysis and the new variations proposed to the HiPPO framework. The experiments are well explained, making the results reproducible.

Minor:
The word sequence is missing in the first sentence of the introduction
 In 2.1, you mention a projection to N-dimensional space, so it may be worthwhile to specify the dimension of the matrices in Equations 2 and 3 for clarity.

P4 an extra bracket appears after the inner product <p,q>

P4 “HiPPO called this” - this sentence seems broken.

Why are there no bold methods in the first free columns in table 1? Isn’t bold representing the leading method?

It would be better to add a conclusion and move one of the figures to the appendix.


**Strength And Weaknesses:**

Strengths: The paper is mostly well-written, and the English level is satisfactory. The paper provides a contribution to an important problem by analyzing a leading method in this field. The analysis also leads to practical improvements in several challenging tasks that require modeling long dependencies with NNs.

Weakness: The paper is very busy with information, without much space for intuitive explanations of some of the results. I would move some plots to the appendix, and add a conclusion and expand on some of the explanations to add intuition to the reader

**Summary Of The Paper:**

The authors study the problem of learning continuous-time dynamical systems using NNs. They focus on the Structured State Space Sequence model (S4) and on a particular matrix for initializing the SSM called a HiPPO matrix. They used Legender polynomials to explain the ability of the HiPPO scheme to capture long dependencies. Their analysis helps them derive new guidelines and variants of the method that lead to performance improvement on Long Range Arena and Path X.

**Summary Of The Review:**

Overall the paper is well-written and provides a nice contribution to the community. The HiPPO framework seems very promising, and the authors provide new analysis and new variations of the methods with Legender polynomials or Fourier Basis functions. This enables them to improve SOTA in several tasks.
For these reasons, I recommend accepting the paper.

---

> ### Author Response · Authors · 2022-11-10
> **Response to Reviewer ZKrd**
>
> We thank the reviewer for their detailed reading of the submission and their support of the paper's merits– “Overall the paper is well-written and provides a nice contribution to the community.”
>
> The reviewer pointed out several areas where the presentation could be improved, which we have addressed in the revision. Thanks for the helpful feedback!
>
> >The paper is very busy with information, without much space for intuitive explanations of some of the results. I would move some plots to the appendix, and add a conclusion and expand on some of the explanations to add intuition to the reader
>
> >It would be better to add a conclusion and move one of the figures to the appendix.
>
> We have moved the earlier Figures 2 and 3 to the appendix and with the freed up space have done the following:
> * Added a new conclusion/discussion section
> * Added more clarifications on which parts of Section 2 are new and added more overview text in Section 3
>
> >The word sequence is missing in the first sentence of the introduction In 2.1, you mention a projection to N-dimensional space, so it may be worthwhile to specify the dimension of the matrices in Equations 2 and 3 for clarity.
>
> We explicitly added in the dimensions of $\mathbf{A,B,C,D}$ in Section 2.
>
> >P4 an extra bracket appears after the inner product <p,q>
>
> >P4 “HiPPO called this” - this sentence seems broken.
>
> >Why are there no bold methods in the first free columns in table 1? Isn’t bold representing the leading method?
>
> All have been fixed - thank you for pointing these out!

---

> > ### Comment · Reviewer_ZKrd · 2022-11-21
> > **After rebuttal**
> >
> > The authors have addressed all my concerns; I keep my score at accept.

---

### Official Review · Reviewer_GLgC · 2022-10-24

**Confidence:** 2
**Correctness:** 4
**Technical Novelty And Significance:** 3
**Empirical Novelty And Significance:** 3
**Recommendation:** 8

**Clarity, Quality, Novelty And Reproducibility:**

The paper is well written but very dense and much information referred to the supplementary. The novelty lies in the application of legendre polynomials and theoretical properties derived including interpretation and initialization of the time-scale parameter \Delta. These derivations are non-trivial and can warrant publication.

**Strength And Weaknesses:**

Strengths:
The paper is in general well written and provides strong theoretical insights into initialization procedures and generalizations of S4.
The approach provides enhanced inference highlighting the utility using exponentially-warped Legendre polynomials.
Insights in regards to the setting of the time-scale parameter \Delta.

Weaknesses:
The paper is an extension of prior work on S4 of Gu et al., 2020; 2021; 2022a and although useful the novelty of the proposed approach appears somewhat limited.

The paper is very dense reverting much information to appendices whereas the paper lacks a discussion and conclusion section and ends somewhat abruptly with the experimental results. I understand the space limitations requires strong compromises but I would encourage the authors to discuss implications at the end to make the paper self-contained within the 9 pages.


**Summary Of The Paper:**

The authors consider theoretical properties of the Structured State Space model (S4) (i.e., a deep learning state space modeling framework) and propose a characterization in terms of exponentially-warped Legendre polynomials demonstrating its theoretical properties and generalizations including to existing S4 procedures thereby in particular enhancing the characterization of long range dependencies. The developed approach is found to be superior on six benchmark from Long Range Arena and naturally extends the work of Gu et al., 2020; 2021; 2022a to address limitations within the existing S4 frameworks.

**Summary Of The Review:**

An extension on the work of S4 including improved initializations and extensions to account for long range dependencies using legendre polynomials. The paper extends existing recent work of Gu et al., 2020; 2021; 2022a addressing limitations of the S4 framework. These results improve upon the existing S4 but the novelty of the framework is somewhat limited, however, the derivations non-trivial and the innovations useful which can warrant publication.

---

> ### Author Response · Authors · 2022-11-10
> **Response to Reviewer GLgC**
>
> We thank the reviewer for taking the time to review our submission. Overall, the reviewer assesses that this work “provides strong theoretical insights into initialization procedures and generalizations of S4” which are non-trivial and well-written. The main weaknesses that the reviewer points out are in the presentation and novelty.
>
> > I would encourage the authors to discuss implications at the end to make the paper self-contained within the 9 pages.
>
> We have updated the paper to have a self-contained conclusion/discussion section, while moving some less important figures (Figures 2 and 3) to the Appendix. We also clarified which parts of Section 2 are new to make the contributions of our framework more transparent.
>
>
> >The paper is an extension of prior work on S4 of Gu et al., 2020; 2021; 2022a and although useful the novelty of the proposed approach appears somewhat limited.
>
> This paper is not just an extension of prior work, but **fills in crucial gaps in the theory of these models** which allows for improvements and new models building on S4. This model is becoming increasingly influential, and our results have already been used to improve on S4 in other ways. Please see our response to reviewer RaMS for a longer discussion on the impact of this work.

---

> > ### Comment · Reviewer_GLgC · 2022-12-05
> > **I thank the authors for addressing my concerns**
> >
> > I thank the authors for clarifying novelty as well as restructuring the paper to make it more self-contained. I have accordingly raise my score to 8.

---

### Official Review · Reviewer_96u9 · 2022-10-27

**Confidence:** 1
**Clarity, Quality, Novelty And Reproducibility:** The paper reads easily. I cannot judg…
**Correctness:** 4
**Technical Novelty And Significance:** 3
**Empirical Novelty And Significance:** 3
**Recommendation:** 6

**Strength And Weaknesses:**

The theoretical contributions of the paper seem interesting. The generalized HiPPO leads to better understanding of the role of initialization in the S4 models. It seems that the experiments can be improved with more baselines.

**Summary Of The Paper:**

The paper makes three key contributions: (1) theoretical understanding of the role of the state matrix in modeling long-term dependencies. (2) derivation of new HiPPO matrices and showing that it captures many of the existing variants. (3) explaining the role of time-scale $\Delta$ in capturing the long-term dependencies.




**Summary Of The Review:**

The paper makes three key contributions: (1) theoretical understanding of the role of the state matrix in modeling long-term dependencies. (2) derivation of new HiPPO matrices and showing that it captures many of the existing variants. (3) explaining the role of time-scale $\Delta$ in capturing the long-term dependencies.

This paper is out of my area of expertise and I cannot judiciously assess the quality of the paper. Unfortunately, I read the paper late and I could not withdraw from reviewing it.

---

> ### Author Response · Authors · 2022-11-10
> **Response to Reviewer 96u9**
>
> We thank the reviewer for taking the time to review our submission. The reviewer agrees that our theory is important and leads to better understanding of S4 models. For a longer discussion on the impact of this work, please see our response to reviewer RaMS.
>
> > It seems that the experiments can be improved with more baselines
>
> This paper focuses specifically on state space models (SSMs), which are state-of-the-art for the datasets we use (in particular, Long Range Arena in Table 1). There are over a dozen Transformer variants which have been tested on LRA (see the [LRA paper](https://arxiv.org/abs/2011.04006) and many follow-ups), which **S4 variants outperform by more than 20 points**. Since other baselines don't perform well and are not the focus of this work, they were not included in Table 1 to reduce clutter.
>
> Additionally, we have now tested more non-Transformer baselines to illustrate that they do not perform well. See response to reviewer RaMS for more details.
>
> |           | ListOps   | CIFAR     | PathX     |
> |-----------|-----------|-----------|-----------|
> |**S4-LegS**| **59.60** | **88.65** | **96.35** |
> |WK-CNN     | 39.95     | 85.01     | 50.00     |
> |LSTM       | 45.10     | 70.15     | 50.00     |

---

### Official Review · Reviewer_RaMS · 2022-10-31

**Confidence:** 3
**Correctness:** 4
**Technical Novelty And Significance:** 2
**Empirical Novelty And Significance:** 2
**Recommendation:** 5

**Clarity, Quality, Novelty And Reproducibility:**

The paper is poorly structured and is not ready for publication in the current form, however the pieces separately are well written (minus the figures). Potentially this is a quality work, but needs work. The work may be of incremental value, but it is difficult to judge.


**Strength And Weaknesses:**

## Strengths
- The paper is a well written exposition of the mathematical details behind the S4 model, although see my comments on the poor structure below.
- Some additional results clarifying previously unjustified parameters of the S4 model.
- Some experimental evidence that the corrections and modifications of S4 guided by the presented theoretical results may lead to improvements in performance.
## Weaknesses
- The paper is poorly structured and at times feels like a part of a chapter on results related to S4 simply pasted into the ICLR format:
  - Discussion and conclusions are missing
  - Figure 1 and Figure 5 are virtually the same and their purpose is not too clear
  - Figure caption of Figure 1 reads like an abstract rather than figure description.
  - Most figures and tables are difficult to interpret - lacking axis labels, subfigures are not addressed in a common way (e.g. Figure 1a ) but with a descriptive textual statements of their locations.
  - As written, the paper is not ready for publication and needs substantial editing and modifications before it can be ready.
- It is unclear whether the results provide enough increment over the S4 paper to warrant a separate publication
  - Understandably, the manuscript consist a lot of background material on S4 and other prior and relevant work. S4 is a relatively recent model and without this background the paper would have been impossible to follow. However, this leads to my doubt of the format these result should be published in. Possibly a note to the original paper would be better.
- Comparisons in Table 1 are lacking 1D CNN and LSTM models
  - I understand that the claim is that these models do not work for long sequences, but a demonstration would be helpful, especially as the S4 model has inspired some boosts in CNN's performance: https://arxiv.org/abs/2206.03398 https://arxiv.org/abs/2210.09298

**Summary Of The Paper:**

The paper provides an intensive mathematical characterization of the S4 model introduced at the last ICLR conference. The paper contains a detailed explanation of the HiPPO initialization schema used in the S4 model and as the main result provides a proof that the seemingly ad-hoc approach of the original S4 is a time-invariant orthogonal SSM. Additionally, the paper provides interpretation of the timescale parameter of S4 and a guidelines on how to set it for a given task.


**Summary Of The Review:**

The paper contains an important explanations of why the S4 model works and how to properly tune it, together with a modification that boosts performance of S4 a little. However, it is not written as a complete self-contained paper and needs improvements to be publishable.

---

> ### Author Response · Authors · 2022-11-10
> **Response to Reviewer RaMS (Part 2)**
>
> Below, we respond to the reviewer's more detailed comments.
>
> ## Presentation Details
>
> We appreciate the reviewer's constructive comments. We note again that an extended version of this submission has been included, as well as notebooks for all figures; we believe that these supplementary materials should address many of the reviewer's concerns. We have also made many improvements to the main 9-page paper following the reviewer's helpful suggestions.
>
> > Discussion and conclusions are missing
>
> A discussion has been added to the revised paper, with an additional longer summary in the extended version of the manuscript.
>
> > Figure 1 and Figure 5 are virtually the same and their purpose is not too clear
>
> We are guessing the reviewer meant Figure 2 and Figure 5 in the original submission. These figures provide visualizations of different HiPPO methods (where Figure 2 are previous HiPPO results, and Figure 5 show our new theory).
>
> However, we agree that they are somewhat redundant, and moreover we emphasize that the animations in the Supplementary are more informative versions of these. We have moved Figure 2 to the Appendix.
>
> > Figure caption of Figure 1 reads like an abstract rather than figure description.
> We have moved Figure 1 to the place where these results are defined (Section 3) and cut down on the caption.
>
> > Most figures and tables are difficult to interpret - lacking axis labels, subfigures are not addressed in a common way (e.g. Figure 1a) but with a descriptive textual statements of their locations.
> * Most of the figures do not have axis labels because they are simply functions; the x-axis is $t$ and the y-axis is the function value. We felt that this is clear from context and that adding labels might clutter the diagrams.
> * Referring to subfigures by position/description instead of adding (a) labels is a stylistic choice, which we think is just as clear and unambiguous.
> * We note again that all figures (and more) are reproducible from the notebooks, which also provide additional context and description.
>
> While we believe that these points are minor, we are open to changing them if the reviewer feels strongly.
>
>
>
> ## Background
> > Understandably, the manuscript consists of a lot of background material on S4 and other prior and relevant work. S4 is a relatively recent model and without this background the paper would have been impossible to follow. However, this leads to my doubt of the format these result should be published in. Possibly a note to the original paper would be better.
>
> We aimed for the paper to be as self-contained as possible and believe that the background is important to understanding the results of this paper.
>
> With that said, we point out that **much of our background material (Section 2) actually does not rehash previous papers, but already provides new framing that simplifies and generalizes previous work** and can be viewed as the start of our contributions.
> * For example, Definition 1 is not introduced in S4 or prior works on deep SSMs, but is a new perspective taken by this work for understanding and visualizing these models.
> * Additionally, all the background on HiPPO (Section 2.2; e.g. Definition 2) is actually a new presentation of the main ideas of HiPPO that simplifies and generalizes the original paper [1].
> * In the revised version, we have added additional text clarifying these contributions and changed the name of Section 2 from "Background" to "Framework".
>
> ## Baselines
>
> This paper focuses specifically on SSM models, and it is also known that standard baselines do not work well on LRA, so they are not included in Table 1. To illustrate, we have taken the same deep neural network architecture and replaced the S4 layers with an LSTM or wide-kernel convolution, with a large hyperparameter sweep, which result in substantially lower performance than S4 on several of the tasks in Table 1.
>
>  |           | ListOps   | CIFAR     | PathX     |
> |-----------|-----------|-----------|-----------|
> |**S4-LegS**| **59.60** | **88.65** | **96.35** |
> |WK-CNN     | 39.95     | 85.01     | 50.00     |
> |LSTM       | 45.10     | 70.15     | 50.00     |
>
>
> The reviewer also singles out the SGConv method [8]. We note that [8] is a concurrent submission and outside the scope of comparison. Moreover, [8] is explicitly modeled after S4, and we believe that its strong performance actually supports the merits of this submission.
> * [8] explicitly designs an exponentially-decaying kernel which was *empirically* designed by examining S4's kernels
> * The fact that S4 produces exponentially-decaying kernels at all is *theoretically* proven by this paper (Corollary 3.3) - as well as the exact rate of decay controllable by $\Delta$.

---

> > ### Author Response · Authors · 2022-11-10
> > **Details of new LRA baselines**
> >
> > Details of LRA baselines:
> > * The "wide-kernel CNN" (WKCNN) is a CNN with a very wide kernel, in the spirit of S4 and the two papers that the reviewer mentioned (CCNN and SGConv). The WK-CNN is a simple version of these where the kernel is freely learnable.
> > * We chose 3 out of 6 of the LRA tasks for a representative sample.
> > * We used the same backbone (e.g. model dimension, residual/normalization structure) as the S4 models from Table 1.
> > * We swept over two model sizes for the baselines. The WK-CNN tried kernel length 512 and 1024; LSTM used hidden size 128 and 256. Resulting models were at least as large as the S4 baselines.
> > * We swept over the learning rate (0.002, 0.004, 0.01) and weight decay parameters (0.01, 0.05). These are a superset of the S4 hyperparameters (LR=0.01, WD=0.05).

---

> ### Author Response · Authors · 2022-11-10
> **Response to Reviewer RaMS**
>
> We sincerely appreciate the reviewer's detailed review and suggestions. The reviewer assesses that the paper contributes important and well-written technical content, and provides constructive feedback about several areas where the paper's presentation and structure can be improved. We first respond to the main points about overall contributions, impact, and presentation, and then address the reviewer's detailed comments.
>
> ## Significance and Impact
>
> > It is unclear whether the results provide enough increment over the S4 paper to warrant a separate publication
>
> * We note that **S4 has become an increasingly influential model**; e.g., there are at least 10 other concurrent submissions to ICLR 2023 that are direct applications or extensions of HiPPO/S4 [5-14]
> * Despite this popularity, **there are gaps in the original papers [1, 2] that are important to resolve** (e.g. lack of mathematical interpretation of S4). This submission introduces the core theory to fill in these gaps, which is non-trivial and is essential to understanding why these methods work.
> * Beyond being an independent contribution to understanding S4, **the ideas in this paper have already been used in downstream work**. For example, several simplifications of S4 [3, 4, 6] are contingent on the theoretical interpretation of HiPPO-LegS (Section 3.1), and also use the theory of $\Delta$ to produce strong results for their experiments [3, 4, 6, 7].
>
>
>
>
>
> ## Presentation
>
> We are grateful for the reviewer's detailed feedback about the presentation and structure of our submission, and have taken much of it into account. Before responding in detail below, we would like to provide a more general comment on the value of this paper.
>
> We believe that the impact of publications to conferences such as ICLR is not limited to the main 9-page paper, but extends to other supplemental aspects of the entire project such as code, datasets, visualizations, etc. We note that beyond the main paper, this submission includes:
> * An extended version of the paper, to be released as a preprint, that unpacks the technical content more carefully, has additional experiments, and includes an extended discussion and conclusion section (added to Supplementary material).
> * Python notebooks that reproduce all figures in the submission, and provide additional animated visualizations that help understand the new theory of this paper.
> * Full code for the new methods and reproducing the experiments will be released as a fork or extension of the original S4 repository.
>
> We hope that the reviewer can take into account these additional materials, which we believe provide a valuable resource for the intended audience of this paper: researchers trying to understand the detailed theory behind S4 and related models.
>
> **References**
>
> [1] Gu et al. "HiPPO: Recurrent Memory with Optimal Polynomial Projections." (NeurIPS 2020)
>
> [2] Gu et al. "Efficiently Modeling Long Sequences with Structured State Spaces." (ICLR 2022)
>
> [3] Gupta et al. "Diagonal State Spaces are as Effective as Structured State Spaces." (NeurIPS 2022)
>
> [4] Gu et al. "On the Parameterization and Initialization of Diagonal State Space Models." (NeurIPS 2022)
>
> [5] "Decision S4: Efficient Sequence-Based RL via State Spaces Layers." Under submission (ICLR 2023)
>
> [6] "Simplified State Space Layers for Sequence Modeling." Under submission (ICLR 2023)
>
> [7] "Liquid Structural State-Space Models." Under submission (ICLR 2023)
>
> [8] "What Makes Convolutional Models Great on Long Sequence Modeling?" Under submission (ICLR 2023)
>
> [9] "Long Range Language Modeling via Gated State Spaces." Under submission (ICLR 2023)
>
> [10] "Deep Latent State Space Models for Time-Series Generation." Under submission (ICLR 2023)
>
> [11] "Spatiotemporal Modeling of Multivariate Signals with Graph Neural Networks and Structured State Space Models." Under submission (ICLR 2023)
>
> [12] "Hungry Hungry Hippos: Towards Language Modeling with State Space Models." Under submission (ICLR 2023)
>
> [13] "Anamnesic Neural Differential Equations with Orthogonal Polynomial Projections." Under submission (ICLR 2023)
>
> [14] "Effectively Modeling Time Series with Simple Discrete State Spaces." Under submission (ICLR 2023)

---

> ### Author Response · Authors · 2022-12-04
> **Gentle Reminder**
>
> We would like to thank you for the constructive feedback again, and give a gentle reminder that the discussion period is closing soon. We would be happy to discuss any further concerns you may have!

---

### Decision · Program_Chairs · 2023-01-20

**Decision:**

Accept: poster

**Justification For Why Not Higher Score:**

 The updated version improves much over the initial submission, and it successfully addressed many concerns raised by the reviewers. I found that there are still quite some small English errors.

**Justification For Why Not Lower Score:**

The paper is in general well written and provides strong theoretical insights into initialization procedures and generalizations of S4.

**Metareview: Summary, Strengths And Weaknesses:**

This paper investigates the problem of learning continuous-time dynamical systems using neural networks, and considers theoretical properties of the Structured State Space model (S4). It provides a characterization in terms of exponentially-warped Legendre polynomials demonstrating its theoretical properties and generalizations. The paper is in general well written and provides strong theoretical insights into initialization procedures and generalizations of S4. The updated version improves much over the initial submission, and it successfully addressed many concerns raised by the reviewers. I found that there are still quite some small English errors.

**Note From Pc:**

if the above contains the word "oral" or "spotlight" please see: "oral" presentation means -> notable-top-5% and "spotlight" means -> notable-top-25%. As stated in our emails, we are disassociating presentation type from AC recommendations